# Synthesis of substituted biphenyls and in vitro evaluation of antimicrobial and anti-biofilm activities supported by in silico ADMET prediction

Saad Alghamdi[1], Ahmed Hassen Shntaif[2]*, Yasser Hussein Issa Mohammed[3]*

1 Department of Clinical Laboratory Sciences, Faculty of Applied Medical Sciences, Umm Al-Qura University, Makkah, Saudi Arabia, 2 Department of Chemistry, College of Science for Women, University of Babylon, Alhilla, Iraq, 3 Department of Pharmacy, Collage of Medicine and Health Science, Hajjah University, Hajjah, Yemen

* wsc.ahmed.hassan@uobabylon.edu.iq (AHS); issayasser16@gmail.com (YHIM)

## Abstract

Antibiotic resistance represents a critical global health challenge, necessitating the development of innovative antimicrobial agents. This research focuses on the design and evaluation of antimicrobial compounds through chemical synthesis and computational methodologies. Biaryl analogues were synthesized via Suzuki coupling reactions and assessed for *in vitro* antimicrobial activity against *Staphylococcus aureus*, *Escherichia coli*, and *Candida albicans*. Computational approaches, including molecular docking, crystal structure optimization, and toxicity profiling, were employed to explore potential molecular interactions and preliminary safety profiles. The synthesized biaryl derivatives demonstrated variable antimicrobial activity across the tested microorganisms. Among these, 2-methoxy-4'-nitro-1,1'-biphenyl (3i) exhibited the most potent antibacterial activity against both Gram-positive and Gram-negative strains. As a preliminary screening, these findings highlight the potential of selected biaryl derivatives as lead candidates for further investigation. Future studies are required to validate activity against resistant strains and to evaluate efficacy *in vivo*.

## Introduction

Antimicrobial resistance (AMR) is a major global health concern that reduces the effectiveness of conventional therapies and contributes to prolonged illness, increased healthcare burden, and mortality [1–3]. In addition to genetic resistance mechanisms, many microorganisms exhibit increased tolerance through biofilm formation, where microbial cells are embedded within a protective extracellular polymeric matrix that limits antimicrobial penetration and promotes persistent infections. Therefore, the discovery of new chemical scaffolds with both antimicrobial and anti-biofilm potential remains an important priority. The bacterium *Staphylococcus*

**Data availability statement:** All relevant data are within the paper.

**Funding:** The author(s) received no specific funding for this work.

**Competing interests:** The authors have declared that no competing interests exist.

aureus (S. aureus) is a Gram-positive pathogenic microorganism responsible for a wide range of clinical infections affecting the skin and soft tissues, lungs, bones, joints, and heart. The emergence of antibiotic resistance and multidrug-resistant strains has made clinical treatment more challenging [4]. In this study, antimicrobial activity was evaluated against clinically relevant microorganisms, including S. aureus (Gram-positive) and Escherichia coli (E. coli) as a representative Gram-negative pathogen commonly associated with urinary tract infections. In addition, Candida albicans (C. albicans) was included as an opportunistic yeast (fungus) that is frequently implicated in mucosal and systemic candidiasis and is well known for its ability to form biofilms, contributing to persistent infections and reduced antimicrobial susceptibility [5]. One of the best practices adopted by medicinal chemists in drug discovery is the use of privileged scaffolds, which can be structurally modified to yield diverse bioactive derivatives [6]. Biphenyl derivatives are widely used as building blocks in medicinal chemistry and have been reported to exhibit various biological activities, including antioxidant and anti-inflammatory properties [7,8]. In addition, biphenyl-containing compounds have been investigated as potential antimicrobial agents, supporting the utility of this scaffold for antimicrobial development [9]. Substituted biphenyls exhibit diverse biological properties due to their structural flexibility and wide scope for substitution, which allows modulation of electronic and physicochemical characteristics [10]. Accordingly, the present study focuses on the synthesis of substituted biphenyl derivatives and the evaluation of their antimicrobial and anti-biofilm activity. In this work, antimicrobial screening and minimum inhibitory concentration (MIC) determinations were conducted to assess the effectiveness of the synthesized compounds against selected microbial strains. Biofilm detection and inhibition were evaluated using methods such as the modified Congo Red Agar (CRAmod) method and the resazurin reduction assay, which are widely applied for phenotypic biofilm analysis and viability-based detection [11,12]. Together, MIC determination and anti-biofilm testing provide a more comprehensive evaluation of antimicrobial performance. To support experimental findings, in silico prediction tools were employed. ADMET properties were estimated to provide preliminary insight into physicochemical and pharmacokinetic behavior that may guide lead optimization [13,14]. Moreover, molecular docking is used to predict plausible binding orientations and interactions between ligands and biological targets, providing qualitative insight into possible molecular mechanisms [15,16]. In this study, docking was performed against E. coli β-ketoacyl-[acyl-carrier-protein] synthase III (FabH), an enzyme involved in bacterial fatty acid biosynthesis and investigated as a promising antibacterial target [17,18]. The docking studies were conducted using the crystal structure PDB ID: 5BNR, and the Molecular Operating Environment (MOE) software was used for docking simulations and visualization [19]. It should be noted that FabH is a bacterial enzyme; therefore, docking results are intended to support antibacterial interpretation only and do not explain antifungal activity against C. albicans. The synthetic route to the target compounds (3a–3k) involves the Suzuki coupling of phenylboronic acid derivatives with appropriately substituted bromobenzene derivatives in ethanol (Scheme 1). Therefore, the objectives of this study were to design and synthesize

substituted biphenyl derivatives and to evaluate their antimicrobial and anti-biofilm potential using experimental and computational approaches.

## Materials and methods

All starting chemicals and materials were obtained from Sigma-Aldrich without purification. For thin-layer chromatography (TLC), Merck precoated TLC plates (silica gel 60 F254) were used. A Thomas Hoover capillary melting point apparatus equipped with a digital thermometer was used to determine the melting points of the synthesized compounds. Infrared (IR) spectra were recorded using the potassium bromide (KBr) pellet method on an FT-IR Shimadzu 8300 spectrophotometer. Analytical scale $^1$HNMR and $^{13}$CNMR spectra were captured in tetramethylsilan (TMS) on a Bruker spectrophotometer. The mass spectra were obtained using a VG70-70H spectrophotometer. Elemental analysis (C, H, N) was carried out using a Perkin-Elmer model 2400 instrument. Melting points were determined by the stuart SMP10 electrothermal digital melting-point apparatus and are uncorrected. Dimethyl sulfoxide (DMSO) was used to dissolve the synthesized compounds. Mueller–Hinton agar (MHA) and Mueller–Hinton broth (MHB) were used for antibacterial assays. For antifungal assays, appropriate standard media were used.

### Procedure for the preparation of substituted biphenyls (3a-k) [12]

Substituted phenyl boronic acid (0.5 g, 3.57 mmol) was added to a solution of (0.72 g, 3.57 mmol) of substituted bromobenzene in a solution of ethanol and water (3:1). The mixture was treated with (2.28 g, 10.72 mmol) of potassium phosphate and degasified under $N_2$ atmosphere for (10–15 min.). After that, (50 mg, 0.036 mmol) of ($K_3PO_4$, Pd (OH)$_2$/C) was added to the mixture, and the reaction was degassed again for (10 min.). The reaction mixture was heated to 65°C for 2–3 h. The progress of the reaction was monitored by thin-layer chromatography (TLC). After the reaction was complete, the mixture was concentrated by evaporation of the solvent. The mixture was cooled to 25°C and diluted with 10 mL of water. The resulting solution was extracted with (2 × 20 mL) $CH_2Cl_2$. The combined organic layer was washed with a saturated NaCl solution and dried over sodium sulfate. The mixture was filtered, and the organic layer was concentrated to yield the crude product. The residue was purified by silica gel column chromatography using hexane as the eluent to afford the corresponding products.

All the compounds synthesized (**3a-d**) were characterized by their $^1$H/$^{13}$C NMR and MS spectra. Compounds **3e–3k** were prepared according to the procedure reported in a previous study. Their structures were confirmed by comparison of their physical properties and spectroscopic characteristics recorded in the present work, which were consistent with those previously reported [12].

**4'-nitro-1,1'-biphenyl (3a).** Yield: 67%, m.p.: 78–80 °C $^1$H NMR (400 MHz, DMSO-$d_6$, ppm): δ 7.20 (t, $J$=8.0 Hz, 1H), 7.35 (t, $J$=8.0 Hz, 1H), 7.51 (d, $J$=8.0 Hz, 1H), 7.56 (d, $J$=8.0 Hz, 1H), 7.80 (d, $J$=8.0 Hz, 2H), 8.41 (d, $J$=8.0 Hz, 2H). $^{13}$C NMR (100 MHz, DMSO-$d_6$, ppm): δ 117.0, 125.2, 126.1, 129.5, 131.2, 144.2, 146.9, 163.2. LC-MS m/z 200 (M + 1). CHN Elemental analysis for $C_{12}H_9NO_2$: Calculated: C, 72.35%; H, 3.85%; N, 7.03%; Found: C, 72.80%; H, 3.55%; N, 7.76%.

**3'-Fluoro-2-methoxy-5'-(trifluoromethyl)-1,1'-biphenyl (3b).** Yield: 81%, m.p.: 124–126 °C, $^1$H NMR (400 MHz, DMSO-$d_6$, ppm): δ 3.76 (s, 3H), 7.23 (t, $J$=8.0 Hz, 2H), 7.28 (d, $J$=8.0 Hz, 1H), 7.31 (s, 1H), 7.37 (d, $J$=8.0 Hz, 2H), 7.52 (d, $J$=8.0 Hz, 1H). $^{13}$C NMR (100 MHz, DMSO-$d_6$, ppm): δ54.8, 113.4, 114.2, 121.6, 125.1, 128.3, 133.4, 142.5, 149.7. LC-MS m/z 287 (M + 1). CHN Elemental analysis for $C_{14}H_{10}F_4O$: Calculated: C, 62.23%; H, 3.73%;; Found: C, 61.97%; H, 3.87%.

**2-Fluoro-2'-methoxy-3-nitrobiphenyl (3c).** Yield: 84%, m.p.: 98–100 °C, $^1$H NMR (400 MHz, DMSO-$d_6$, ppm): δ 3.65 (s, 3H), 6.89 (d, $J$=8.0 Hz, 1H),7.35 (t, $J$=8.0 Hz, 2H), 7.37 (d, $J$=8.0 Hz, 1H), 7.40 (d, $J$=8.0 Hz, 1H), 7.77 (d, $J$=8.0 Hz, 1H), 8.11 (d, $J$=8.0 Hz, 1H). $^{13}$C NMR (100 MHz, DMSO-$d_6$, ppm): δ54.9, 114.5, 121.4, 124.1, 128.6, 134.7, 155.3, 162.5. LC-MS m/z 248 (M + 1). CHN Elemental analysis for $C_{13}H_{10}FNO_3$: Calculated: C, 63.16%; H, 4.08%; N, 5.67; Found: C, 62.97%; H, 3.97%: N, 5.78.

**2'-chloro-2-fluoro-5-methoxy-1,1'-biphenyl (3d).** Yield: 77%, m.p.: 145−147 °C, $^1$H NMR (400 MHz, DMSO-$d_6$, ppm): δ 3.84 (s, 3H), δ 7.21–7.39 (s, 5H), 7.69 (s, 1H), 7.21 (t, $J=8.0$ Hz, 2H), 7.34 (d, $J=8.0$ Hz, 1H). $^{13}$C NMR (100 MHz, DMSO-$d_6$, ppm): δ56.19, 113.82, 114.17, 117.90, 128.5, 130.0, 133.06, 137.22, 151.11, 156.12. LC-MS m/z 238 (M+1). CHN Elemental analysis for $C_{13}H_{10}ClFO$: Calculated: C, 65.97%; H, 4.26%; Found: C, 65.72%; H, 4.52%.

**2,4'-Difluoro-3'-(trifluoromethyl)biphenyl (3e).** Yield 85%, m.p. 140–143 °C. $^1$H NMR (400 MHz, DMSO-$d_6$, ppm): δ 7.23 (t, $J=8.0$ Hz, 2H), 7.45 (t, $J=8.0$ Hz, 1H), 7.68 (d, $J=8.0$ Hz, 2H), 7.74 (d, $J=8.0$ Hz, 1H), 7.84 (s, 1H).

**2,4'-Difluoro-3'-(trifluoromethoxy)biphenyl (3f).** Yield 70%, m.p. 134–136 °C. $^1$H NMR (400 MHz, DMSO-$d_6$, ppm): δ 7.26 (t, $J=8.0$ Hz, 2H), 7.45 (d, $J=8.0$ Hz, 2H),7.52 (d, $J=8.0$ Hz, 2H), 7.49 (s, 1H).

**2,2'-Difluoro-3-nitrobiphenyl (3g).** Yield 70%, m.p. 170–172 °C. $^1$H NMR (400 MHz, DMSO-$d_6$, ppm): δ 7.21 (d, $J=8.0$ Hz, 1H), 7.32 (t, $J=8.0$ Hz, 2H), 7.48 (d, $J=8.0$ Hz, 2H), 7.72 (d, $J=8.0$ Hz, 1H), 8.15 (d, $J=8.0$ Hz, 1H).

**2,2'-Difluoro-5-methoxybiphenyl (3h).** Yield 90%, m.p. 110–112 °C. $^1$H NMR (400 MHz, DMSO-$d_6$, ppm): δ 3.74 (s, 3H), 6.71 (d, $J=8.0$ Hz, 1H), 6.92 (d, $J=8.0$ Hz, 2H), 7.20 (t, $J=8.0$ Hz, 2H), 7.47 (d, $J=8.0$ Hz, 1H),

**2-methoxy-4'-nitro-1,1'-biphenyl (3i).** Yield 76% m.p. 139–140 °C. $^1$H NMR (400 MHz, DMSO-$d_6$, ppm): δ 3.71 (s, 3H), 7.15 (t, $J=8.0$ Hz, 1H), 7.28 (t, $J=8.0$ Hz, 1H), 7.35 (d, $J=8.0$ Hz, 1H), 7.42 (d, $J=8.0$ Hz, 1H). 7.67 (d, $J=8.0$ Hz, 2H), 8.16 (d, $J=8.0$ Hz, 2H).

**4'-Fluoro-2-methoxy-3'-(trifluoromethyl)biphenyl (3j).** Yield 82% m.p. 149–152 °C. $^1$H NMR (400 MHz, DMSO-$d_6$, ppm): δ 3.81 (s, 3H), 7.13 (t, $J=8.0$ Hz, 1H), 7.22 (t, $J=8.0$ Hz, 2H), 7.39 (d, $J=8.0$ Hz, 1H), 7.48 (d, $J=8.0$ Hz, 1H). 7.81 (d, $J=8.0$ Hz, 2H), 7.95 (d, $J=8.0$ Hz, 1H).

**2-Fluoro-2',5-dimethoxybiphenyl (3k).** Yield 82% m.p. 115–116 °C. $^1$H NMR (400 MHz, DMSO-$d_6$, ppm): δ 3.69 (s, 6H), 6.78 (d, $J=8.0$ Hz, 1H), 6.97 (s, 1H), 7.11 (t, $J=8.0$ Hz, 2H), 7.23 (d, $J=8.0$ Hz, 1H), 7.51 (d, $J=8.0$ Hz, 1H), 7.61 (d, $J=8.0$ Hz, 1H).

### *In silico* assays

Molecular docking studies were carried out to better understand the synthesized compounds and the protein [17] to predict the orientation and conformation of the ligands in the protein's active site and to determine the binding affinity. Molecular docking studies were performed with MOE software to investigate possible interactions of the most potent ligands. The PDB file for *E. coli* Fabh (PDB ID: 5bnr) was downloaded from the Protein Data Bank (http://www.rcsb.org) [17,9].

*In silico* prediction. ADMET properties of the synthesized compounds were assessed using SwissADME, AdmetSAR, and pkCSM online software to calculate physicochemical and pharmacological properties for four selected molecules [5,14].

### Biological assays

**Bacterial isolates and antimicrobial activity assays.** The *S. aureus* and *E. coli* isolates used in this study were obtained from the Microbiology Laboratory, College of Applied Sciences, University of Hajjah, and originated from routine diagnostic urine cultures. Experiments were conducted on archived, de-identified isolates after completion of routine clinical testing; no patient identifiers or private clinical data were accessed, and results were not used to guide diagnosis, treatment, or clinical decision-making; therefore, patient clinical outcomes were not affected. Ethical approval/waiver was obtained from the University of Hajjah. The isolates were previously identified by phenotypic and biochemical methods and preserved for research use [19]. For assays, each isolate was cultured separately in tryptic soy broth at 37 °C for 24 h, harvested by centrifugation (5000 × g, 10 min) [18], washed twice with sterile phosphate-buffered saline (PBS), and resuspended in sterile 1.0% NaCl to a final density of approximately $1.8 \times 10^8$ CFU/ml [20]. Antimicrobial resistance profiling (e.g., MRSA/ESBL status) was not performed within the scope of this study and is acknowledged as a limitation; future work will include expanded testing against a broader panel of well-characterized multidrug-resistant clinical isolates.

### *In vitro* evaluation of the antimicrobial activity of synthesized compounds against gram-positive and gram-negative bacteria

Two different bacterial strains (*Staphylococcus aureus* and *Escherichia coli*) and one yeast strain (*Candida albicans*) were used to evaluate the in vitro antimicrobial activity of the synthesized compounds. *S. aureus* (Gram-positive) and *E. coli* (Gram-negative) were selected to provide an initial comparison across organisms with different cell-envelope structures, while *C. albicans* was included as a fungal (yeast) species for supportive screening. Bacterial inoculum was standardized to 0.5 McFarland in sterile saline, and Mueller–Hinton agar (MHA) plates were inoculated by evenly swabbing the surface with a sterile cotton swab. Wells of 6 mm diameter were punched into the agar, and 100 µl of each compound solution (prepared in DMSO) was added to the wells; DMSO was used as the standard solvent for compound dissolution and served as the vehicle (solvent) control at the same final concentration in all assays [21,22]. Plates were incubated at 37 °C for 24 h, and antimicrobial activity was assessed by measuring inhibition zone diameters (mm) around the wells. These experiments provide preliminary comparative screening of antimicrobial activity across Gram-positive bacteria, Gram-negative bacteria, and a yeast strain under standardized in vitro conditions.

### Determination of antibacterial inhibitory concentration by agar well diffusion

The antibacterial activity of the synthesized compounds was evaluated using the agar well diffusion method. A diffusion-based inhibitory concentration was defined as the lowest tested concentration that produced a measurable inhibition zone. DMSO (same final concentration as treatments) served as the vehicle control [23,24]. Compounds were screened at 100 µg/ml (w/v in DMSO) and tested as ten-fold serial dilutions (100, 10, 1, 0.1, 0.01 µg/ml). Briefly, 1.0 mL of $1.8 \times 10^8$ CFU/ml was added to sterile Petri dishes; molten Mueller–Hinton agar was poured, mixed gently, and allowed to solidify. Four wells per plate were punched using a sterile 6 mm cork borer, and 100 µl of each concentration was added to the wells. Plates were incubated at 37 °C for 24 h, and inhibition zone diameters (mm) were measured.

### Antimicrobial potency of compound 3i: MIC by resazurin broth microdilution

The minimum inhibitory concentration (MIC) was determined for the most active compound 3i using a broth microdilution assay in accordance with standardized antimicrobial susceptibility testing principles (CLSI M07 and ISO 20776−1) [25,26]. Resazurin was used as a viability indicator; metabolically active cells reduce the blue, non-fluorescent dye to a pink fluorescent product, allowing growth inhibition to be detected by colour change [27]. The MIC was defined as the lowest concentration showing no visible growth/no resazurin colour change compared with the growth control. This resazurin-based microdilution approach has been widely reported as a sensitive and reproducible method for MIC determination [28,29].

### Evaluation of the Biofilm formation and inhibition assays (CRAmod and microtiter plate quantification

The modified Congo red agar method (CRAmod) was used as a qualitative assay to screen for the biofilm-forming phenotype and to assess the effect of compound 3i on biofilm development [25]. For quantitative measurement of biofilm biomass, a 96-well polystyrene microtiter plate assay was performed. Cultures were grown in TSB at 37 °C for 24 h and inoculated into wells containing TSB supplemented with 1% (w/v) xylose as a defined carbohydrate supplement to promote consistent and reproducible biofilm formation under static conditions (applied equally to treated and control wells). After incubation, wells were washed with PBS to remove non-adherent cells, stained with 0.1% safranin, and absorbance was measured at 492 nm (OD492) using a Multiskan GO microplate reader [26]. Biofilm formation was additionally evaluated on glass surfaces under the same growth conditions using a 5 ml culture volume (instead of 200 µl in microtiter plates) [30]. Results from these assays were used to assess the inhibitory effect of compound 3i on biofilm formation in the tested microorganisms.

## Results and discussion

Suzuki coupling reactions were employed to synthesize biphenyl compounds (3a–k) by reacting substituted bromobenzene with substituted phenylboronic acids in a 1:1 ratio, using $Pd(OH)_2$ as the catalyst and potassium phosphate as the base, at 65 °C. The synthesized compounds were thoroughly characterized by spectroscopic techniques such as IR and ¹H NMR, and the resulting data were in excellent agreement with the calculated values for the proposed structures. Furthermore, preliminary testing revealed promising antimicrobial activity in the newly synthesized compounds.

### *In silico* toxicity studies

*In silico* toxicity assessments utilize computational methods to predict the potential toxicity of a chemical compound. They can be used for drug development and assessing the safety of substances contacting humans or the environment. These assessments are a valuable tool for predicting and evaluating the potential risks of chemical substances.

### Comprehensive assessment of the ADMET properties of compound 3i and streptomycin

The Swiss ADME, Admet SAR, and pkCSM software were used to predict the pharmacokinetic properties of the 3i compound (table 1). Lipinski's descriptors evaluate molecular properties related to drug pharmacokinetics in humans, especially oral absorption. Compound 3i did not violate Lipinski's rule of five. In terms of its molecular properties, compound 3i exhibits moderate lipophilicity, moderate to high polarity, and hydrogen bonding potential. The compound also shows moderate absorption, availability, and solubility. The drug similarity model score is 1.33, indicating that the evaluated compound has a high degree of drug similarity. This suggests that compound 3i likely possesses drug-like properties based on its physicochemical and structural characteristics. Further studies may be needed to assess its potential as a drug candidate. Additionally, computational analysis revealed relatively low water solubility (−4.449 mol/l), moderate Caco-2 intestinal permeability (0.6708 cm/s), and high gastrointestinal (GI) absorption (94.592%), with predicted GI absorption surpassing that of streptomycin. In silico predictions also show that 3i has good permeability through the blood–brain barrier (BBB) (4.52 log BB), high permeability in the central nervous system (0.3062 log PS), and low skin permeability (−2.65 log Kp), similar to streptomycin. Compound 3i is likely biologically active against the listed protein targets. These findings add to ongoing research aimed at identifying safe and effective drug candidates for various diseases and conditions. The

Table 1. The key ADMET characteristics of the compound 3i.

| Properties | Molecule | |
|---|---|---|
| | Compound 3i | streptomycin |
| Formula | $C_{13}H_{11}NO_3$ | $C_{21}H_{39}N_7O_{12}$ |
| MW | 229.24 | 581.57 |
| Num. Heavy atoms | 17 | 40 |
| Num. Aromatic heavy atoms | 12 | 0 |
| Num. Rotatable bonds | 3 | 9 |
| Num. H-bond acceptors | 3 | 15 |
| Num. H-bond donors | – | 12 |
| MR | 67.19 | 130.43 |
| TPSA | 55.05 Å² | 336.43 |
| Consensus Log P | 2.48 | −5.84 |
| Ali Class | Moderately soluble | Highly soluble |
| Silicos-IT Log S | −4.47 | 4.60 |
| log Kp | −5.12 | −15.52 |
| Lipinski #violations | Yes; 0 violation | No; 3 violations: MW > 500, N or O > 10, NH or OH > 5 |

compound was identified as a substrate or inhibitor of P-glycoprotein I, indicating that its transport and excretion depend on the P-glycoprotein pathway. In silico results also suggest that compound 3i has limited permeability based on the computed skin permeability coefficient (log Kp = −5.12) (Table 2) (Fig 1).

In silico analysis predicted that both 3i and streptomycin would not be re-absorbed by renal OCT2 (OCT2 substrates), which is important for the elimination of cationic molecules. The in silico estimation suggested that 3i is excreted moderately through the kidneys (0.335 log mL/min/kg), with a low steady-state volume of distribution (VDss), indicating moderate distribution that results in higher concentrations in tissues than in plasma (Table 3).

**Table 2. *In silico* ADMET and toxicity prediction for compound 3i compared with streptomycin.**

| Parameter | Compound 3i | Streptomycin |
|---|---|---|
| Water solubility (log mol/L) | −4.449 | −2.892 |
| Blood–Brain Barrier (BBB) | BBB+ | BBB− |
| BBB **score** (*not probability*) | 4.52 | 0.9712 |
| Caco-2 permeability | Caco2+ | Caco2− |
| Caco-2 probability | 0.6708 | 0.6968 |
| Human intestinal absorption (% absorbed) | 94.592 | 0 |
| CYP inhibitory promiscuity | High | Low |
| CYP inhibition probability | 0.3062 | 0.8818 |
| CNS permeability (log PS) | −2.65 | −2.735 |
| Acute oral toxicity class | II | IV |
| Acute toxicity probability | 0.4839 | 0.6171 |
| Rat oral LD50 (mol/kg) | 1.2618 | 1.8409 |
| P-gp inhibitor I | Yes | No |
| P-gp inhibitor II | No | No |
| P-gp substrate | No | Yes |

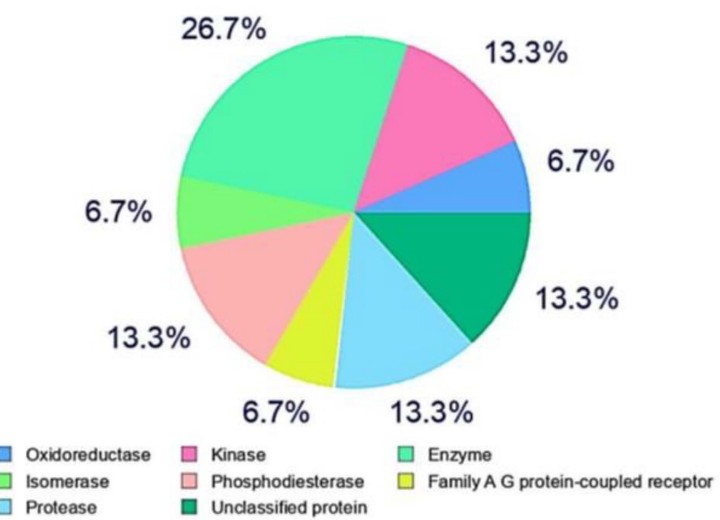

**Fig 1. Probability for the compound 3i assumed as bioactive to the above protein as target.**

**Table 3. The in silico pharmacological excretion characteristics of KTU-286 predicted.**

| Pharmacological Test | 3i compound | streptomycin |
|---|---|---|
| VDss (log L/Kg) | 0.268 | −0.274 |
| Total renal clearance (log mL/min/Kg) | 0.335 | 0.009 |
| Renal OCT2 substrate | No | No |

Although these *in silico* predictions are promising, they need to be confirmed through experimental validation. Additional studies, such as cytotoxicity tests, pharmacokinetic analysis, and efficacy assessments, are essential to confirm the therapeutic potential of compound 3i.

## Docking results

**Molecular docking analysis of proposed compounds.** Molecular docking was performed using the Molecular Operating Environment (MOE) to investigate possible binding interactions of the synthesized compounds with the antibacterial target β-ketoacyl-[acyl-carrier-protein] synthase III (FabH) from E. coli. The crystal structure of FabH was retrieved from the Protein Data Bank (PDB ID: 5BNR) and prepared in MOE by adding hydrogen atoms, assigning appropriate protonation states, and minimizing the structure using the Smart Minimizer algorithm. The binding site was defined based on the position of the co-crystallized ligand. To validate the docking protocol, the co-crystallized ligand was extracted and re-docked into the FabH active site using the same docking settings. The predicted pose was compared with the experimental orientation, and the protocol was considered acceptable when the RMSD was 1.36 Å. After validation, the synthesized compounds (3a–3k) were docked into the same binding pocket, and the best-ranked poses were selected based on the docking score (more negative values indicate more favorable predicted binding) and the plausibility of key binding interactions. The predicted binding mode of the most active compound 3i is shown in Fig 2.

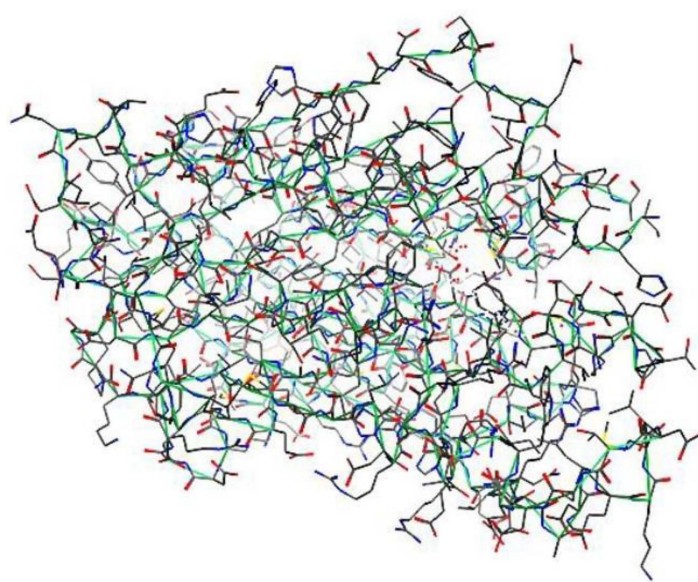

**Fig 2. The protein structure model of *E. coli* Fabh.**

**The physicochemical attributes of 5bnr protein.** The crystal structure of *E. coli* β-ketoacyl-[acyl-carrier-protein] synthase III (FabH) was retrieved from the Protein Data Bank (PDB ID: 5BNR) (Figs 3 and 4). The physicochemical and crystallographic data, including the space group, resolution, molecular weight, chain identifier, and number of amino acid residues, were obtained from the PDB and summarized in Table 4. Before molecular docking, the protein structure was prepared to enhance structural stability. This process involved correcting bond orders and formal charges, adding missing hydrogen atoms, and adjusting incomplete side chains and terminal groups. The structure was energy-minimized using an appropriate force field to remove steric clashes and optimize local geometry, including the orientation of side-chain

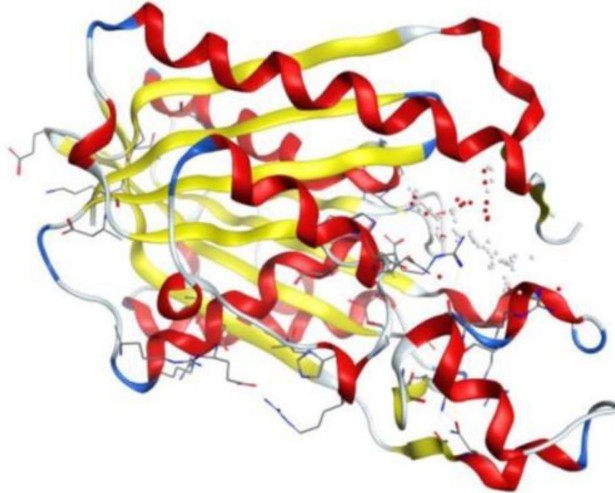

**Fig 3. Crystal structure of *E. coli* Fabh protein from (PDB ID: 5BNR) in ribbon models.**

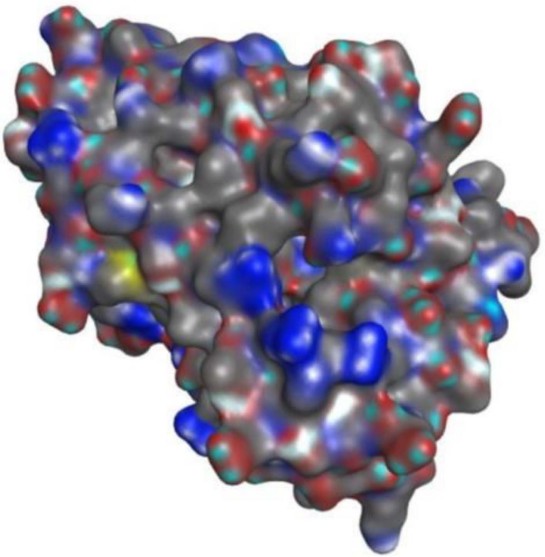

**Fig 4. Crystal structure of E. coli Fabh protein from (PDB ID: 5BNR) in surface and maps models.**

**Table 4. The physicochemical attributes of 5bnr have been obtained from the PDB repository.**

| Protein 5BNR: From Amino-Acid Chains to Physiochemical Properties | |
| --- | --- |
| Parameters | Value |
| Cell Space Group | P $4_1$ $2_1$ 2 |
| Crystallographic Resolution | 1.89 Å |
| Molecular Weight | 33.88 kDa |
| Amino-acid Chain Name | A |
| Number of Amino-acid Residues | 309 |

hydroxyl groups. Minimization was carried out under a positional restraint protocol, limiting deviations from the original crystal coordinates with an RMSD tolerance of 0.3 Å.

**Computational validation in digital modeling.** In this research, we used various software tools and algorithms to analyze the interaction between the *E. coli* Fabh protein and synthesized compounds. The validation of the protein using MOE software ensured the accuracy of the results, and the prediction of the ligand core structures using ChemDraw Professional 16.0 provided insight into the molecular composition of the compounds. The optimization and minimization of the energy of the ligands' 3D structures using Chem3D 16.0 software enabled more accurate prediction of potential interactions between the ligands and the target protein. Docking studies were conducted in MOE, with protein-ligand interactions performed within the same program. The protein was imported as a ligand-free receptor, and the missing bond orders, hybridization, and charges were assigned to it, which was already implemented in the program. The cavity detection algorithm was used to identify potential binding sites, and docking was performed with 11 predicted ligands. A cavity docking procedure with an energy grid resolution of 0.45 Å was used, taking the default parameters. The algorithm Mol Dock SE was used, with a total of 1 as the best run, a population size of 50, and a maximum of 1500 interactions. MOE Virtual Docker was used to visualize the output. The scores of active compounds in Scheme 1 were summarized, and the fitness scores for each ligand in the 5bnr receptor were presented in Table 5 and Fig 5. When compared with the standard (streptomycin) (docking score −7.23) and the most active synthesized compound 3i (docking score −6.48), the docking results suggest that several compounds in Scheme 1 exhibit reasonable predicted binding within the active site of the target protein. Overall, the docking scores of compounds 3a–3k were within a comparable range, indicating similar binding modes under the applied scoring function. Although streptomycin produced the most favorable docking score, compound 3i showed one of the best docking performances among the synthesized derivatives. This may be attributed to favorable electrostatic contributions and the formation of stabilizing interactions, including hydrogen bonding,

3a:$R_1$=H, $R_2$=H, $R_3$=$NO_2$, $R_4$=H, R'=H
3b:$R_1$=H, $R_2$=$CF_3$, $R_3$=H, $R_4$=H, R'=$OCH_3$
3c:$R_1$=F, $R_2$=F, $R_3$=H, $R_4$=H, R'=$OCH_3$
3d:$R_1$=F, $R_2$=H, $R_3$=H, $R_4$=$OCH_3$, R'=Cl
3e:$R_1$=H, $R_2$=$CF_3$, $R_3$=F, $R_4$=H, R'=F

3f:$R_1$=H, $R_2$=$OCF_3$, $R_3$=F, $R_4$=H, R'=F
3g:$R_1$=H, $R_2$=H, $R_3$=$NO_2$, $R_4$=F, R'=F
3h:$R_1$=F, $R_2$=H, $R_3$=H, $R_4$=$OCH_3$, R'=F
3i: $R_1$=H, $R_2$=H, $R_3$=$NO_2$, $R_4$=H, R'=$OCH_3$
3j:$R_1$=H, $R_2$=$CF_3$, $R_3$=F, $R_4$=H, R'=$OCH_3$
3k:$R_1$=F, $R_2$=H, $R_3$=H, $R_4$=$OCH_3$, R'=$OCH_3$

**Scheme 1. Synthetic route to target compounds (3a-k) using the Suzuki-Miyaura reaction.**

**Table 5. Molecular docking scores of compounds 3a–3k against *E. coli* FabH (PDB ID: 5BNR).**

| Mol | rseq | mseq | s | Rmsd_refine | E_conf | E_place | E_score 1 | E_refine | E_score 2 |
|---|---|---|---|---|---|---|---|---|---|
| **3i** | 1 | 1 | −6.4004 | 1.5551 | 47.8033 | −63.8836 | −9.6765 | −31.6765 | −6.4004 |
| Strp | 1 | 12 | −7.2315 | 2.3775 | −337.197 | −51.1887 | −9.9224 | −40.2174 | −7.2315 |
| 3a | 1 | 6 | −5.8467 | 0.9989 | 29.8354 | −75.2196 | −10.2327 | −27.9491 | −5.8467 |
| 3b | 1 | 8 | −6.1866 | 1.5351 | 29.3389 | −53.6479 | −10.2326 | −26.7255 | −6.1866 |
| 3c | 1 | 9 | −6.0612 | 1.0396 | 30.7571 | −50.8721 | −9.6401 | −23.5591 | −6.0612 |
| 3d | 1 | 11 | −5.9582 | 1.3471 | 21.7551 | −47.2196 | −8.7493 | −25.6293 | −5.9582 |
| 3e | 1 | 2 | −6.0586 | 1.3013 | 23.6346 | −48.4103 | −9.0213 | −27.5857 | −6.0586 |
| 3f | 1 | 3 | −6.0580 | 0.9236 | 32.5269 | −60.4496 | −9.7404 | −29.5206 | −6.0580 |
| 3g | 1 | 4 | −5.8510 | 1.5990 | 29.8380 | −64.4290 | −9.5882 | −27.9510 | −5.8510 |
| 3h | 1 | 5 | −5.9537 | 0.5688 | 52.4212 | −56.3914 | −9.6838 | −28.3329 | −5.9537 |

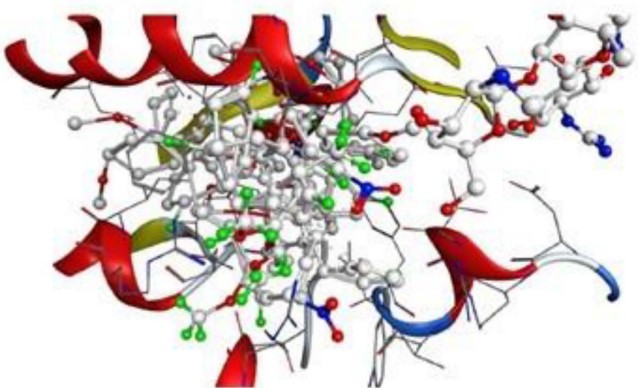

**Fig 5. Interactions of compounds (3a-3k) in cavity of 5BNR protein shown as backbone structure with labeled residuals and Ligands are in ball-stick model.**

with key active-site residues. Detailed analysis of the predicted binding mode revealed that compound 3i is stabilized by two key hydrogen bonding interactions (Fig 6). The first interaction occurs between a carbonyl oxygen of compound 3i and the NH1 group of Gly 209 at a distance of 2.77 Å, contributing −6.7 kcal/mol to the binding energy. The second hydrogen bond is formed between an oxygen atom (likely from a hydroxyl or carbonyl group) of the compound and the backbone nitrogen of Asn 274, with a bond distance of 2.90 Å and an interaction energy of −5.0 kcal/mol. It should be noted that docking scores provide qualitative insight into potential binding orientations and interactions, and do not necessarily correlate directly with experimental antimicrobial potency as observed in Table 6. This approach facilitated the identification of potential candidate compounds for the development of new antibacterial agents, with compound 3i demonstrating promising inhibition and binding affinity toward the *E. coli* FabH protein.

**Structure–activity relationship (SAR) analysis.** The antimicrobial screening results across the synthesized biaryl series (3a–3k) indicate that both electronic effects and the balance of lipophilicity and polarity contribute to antimicrobial performance. A clear trend was observed for the nitro-containing derivatives. Compounds **3a** (R3 = NO₂, R′ = H) and **3i** (R3 = NO₂, R′ = OCH₃) share the same nitro substitution on the biaryl core; however, **3i** showed stronger antibacterial activity. This suggests that introducing a methoxy group on the second aryl ring (R′ = OCH₃) improves activity, likely by enhancing the overall physicochemical profile (e.g., improved partitioning into bacterial membranes and/or more favorable

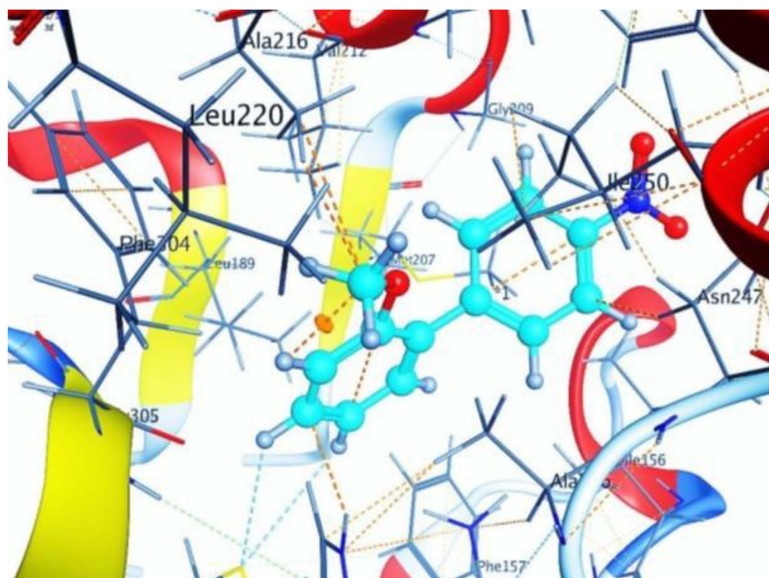

**Fig 6. Hydrogen bonding interaction of compound 3i with protein 5BNR shown in blue ball-stick model.**

**Table 6. Anti-microbial activity of the tested compounds. Inhibition zones are being expressed as the mean±SD (mm) n=3. *Staphylococcus aureus, E. coli* and *C. albicans*.**

| Substance code | Inhibition zone by mm | | |
|---|---|---|---|
| | *S. aureus* | *E. coli* | *C. albicans* |
| 3a | NZ | NZ | NZ |
| 3b | NZ | NZ | NZ |
| 3c | 18.11±1.01 | 20.21±2.01 | 7.16±0.17 |
| 3d | NZ | NZ | NZ |
| 3e | 3.11±0.02 | 7.11±2.12 | NZ |
| 3f | 12.31±3.01 | 14.21±1.30 | NZ |
| 3g | 4.31±1.22 | 4.11±0.12 | NZ |
| 3h | NZ | NZ | NZ |
| **3i** | 19.33±2.13 | 20.11±1.11 | 13.13±0.01 |
| 3j | NZ | NZ | NZ |
| 3k | 8.28±1.30 | 7.22±1.21 | 5.02±2.25 |
| Positive control (streptomycin) | 20.01±1.22 | 17.02±0.21 | 13.90±1.11 |
| Fluconazole | NZ | NZ | 18.31±2.11 |
| (DMSO) | NZ | NZ | NZ |

NZ: no Inhibition zone.

interaction geometry at the binding site). In addition, the p-nitro group (R3=NO$_2$) may enhance activity by increasing electron-withdrawing character and providing strong hydrogen-bond acceptor capacity, thereby supporting stabilizing electrostatic and hydrogen-bonding interactions with biological targets.

Fluorination produced strain-dependent effects. The difluoro analogue **3c** (R1=F, R2=F, R′=OCH$_3$) displayed improved activity, particularly against *C. albicans*, compared with many non-fluorinated derivatives, suggesting that fluorination can

enhance antifungal performance, potentially by increasing lipophilicity and promoting interaction with fungal membranes or improving binding complementarity. In contrast, fluorinated analogues containing strongly lipophilic groups such as $CF_3$ or $OCF_3$ (e.g., 3b, 3e, 3f, 3j) showed variable antibacterial activity, which may reflect competing effects: increased hydrophobicity can support membrane penetration, but excessive fluorination can reduce aqueous solubility and agar diffusion, leading to weaker apparent activity in diffusion-based assays.

Substitution at $R4 = OCH_3$ also appeared to modulate activity in some cases (e.g., 3d and 3h), suggesting that electron-donating substituents on the aryl ring may influence target interaction patterns or overall polarity. Overall, the preliminary SAR indicates that the best antibacterial performance in this series is achieved with a nitro group on the biaryl scaffold and a methoxy substituent on the second ring (**3i**), whereas fluorination may contribute more prominently to antifungal activity (e.g., **3c**). These findings provide a basis for future optimization, including testing against a broader strain panel (including resistant isolates), confirming MICs by broth microdilution, and exploring mechanistic targets experimentally.

**Anti-microbial activity.** The antimicrobial activity of the synthesized compounds was evaluated against two bacterial strains (*Staphylococcus aureus* and *Escherichia coli*) and one fungal strain (*Candida albicans*) using the agar well diffusion method. The activity was assessed by measuring the inhibition zone diameter (mm), and the results are presented as the mean ± SD (Table 6 and Fig 7). Overall, the compounds exhibited variable antimicrobial effects across microorganisms. Compound **3i** demonstrated the highest antibacterial activity against both *S. aureus* and *E. coli*, indicating strong broad-spectrum potential against Gram-positive and Gram-negative bacteria. In contrast, compound **3c** showed the best antifungal activity against *C. albicans*. Moderate antibacterial activity against *S. aureus* was observed for compounds **3b**, **3d**, **3e**, and **3h**, while moderate activity against *E. coli* was observed for compounds **3c**, **3i**, and **3j**. For *C. albicans*, compounds **3i** and **3j** produced moderate inhibition zones. Compounds **3a**, **3g**, and **3k** showed no detectable antimicrobial activity against any of the tested microorganisms under the experimental conditions used.

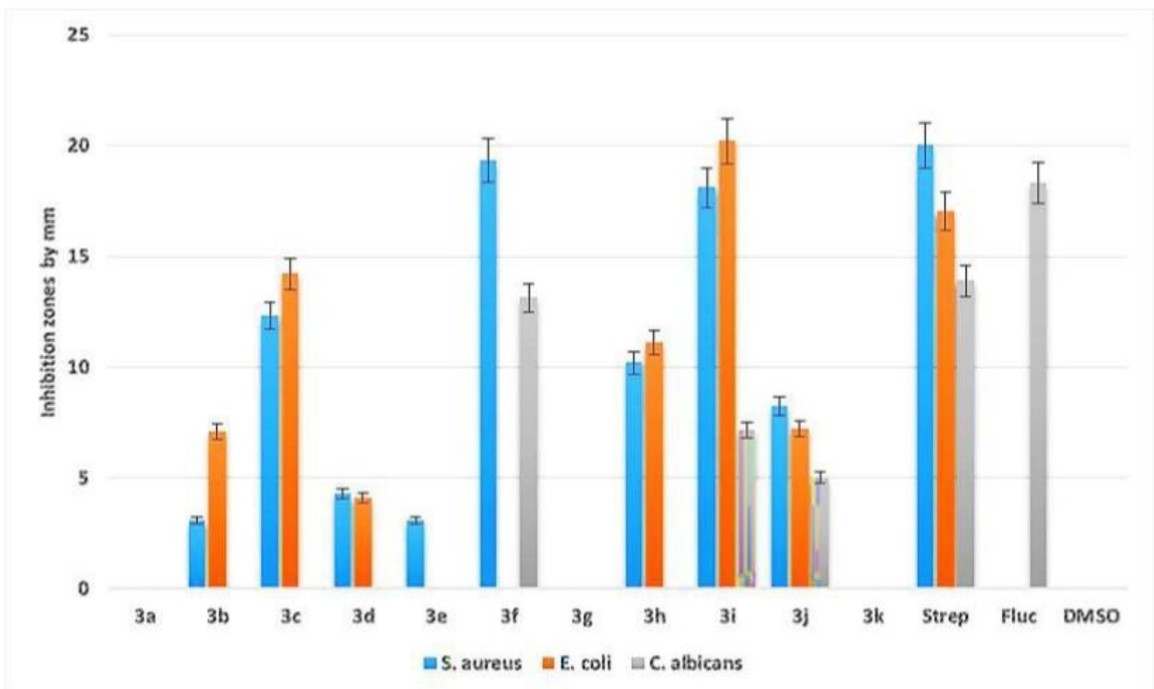

**Fig 7. Sensitivity test of the synthesized compounds against three clinical isolates. Inhibition zones are expressed in mm.**

Appropriate controls were included to validate the assays. DMSO was used as the vehicle (solvent) control at the same final concentration applied in the treated samples, confirming that the solvent did not contribute to microbial inhibition. Streptomycin served as the positive control for antibacterial activity, producing clear inhibition zones against *S. aureus* and *E. coli*. Fluconazole was used as the positive control for antifungal activity and showed inhibition against *C. albicans*, as expected, while exhibiting no inhibitory effect on the bacterial strains. In addition to the preliminary screening (Table 6), further dilution-based testing was performed to compare inhibitory concentrations of the most active compounds and to confirm sensitivity (Table 7 and Fig 8). Consistent with the initial screening, compound **3i** remained the most active, while several other compounds showed moderate activity, and others were inactive. To further estimate potency, diffusion-based screening was performed using the agar well diffusion assay, and the highest inhibitory concentration was defined as the lowest tested concentration that produced a visible inhibition zone (Table 7). Among all compounds, **3i** showed the strongest activity, yielding the lowest agar well diffusion test result values against *S. aureus* (5.2 µg/ml) and *E. coli* (3.4 µg/ml). Compound **3f** showed moderate activity with values of 14.6 µg/mL against *S. aureus* and 21.4 µg/mL against *E. coli*.

**Table 7. Diffusion-based inhibitory concentration of the synthesized compounds against the three tested clinical isolates.**

| Substance code | Diffusion-based inhibitory concentration by µg/ml | | |
| --- | --- | --- | --- |
| | *S. aureus* | *E. coli* | *C. albicans* |
| 3a | NC | NC | NC |
| 3b | NC | NC | NC |
| 3c | 33.1 | 28.9 | 45.5 |
| 3d | NC | NC | NC |
| 3e | 25.5 | 23.2 | NC |
| **3f** | 14.6 | 21.4 | NC |
| 3g | 33.9 | 43.3 | NC |
| 3h | NC | NC | NC |
| **3i** | **5.2** | **3.4** | **8.3** |
| 3j | NC | NC | NC |
| 3k | 43.2 | 45.5 | 42.1 |

NC: none Inhibition.

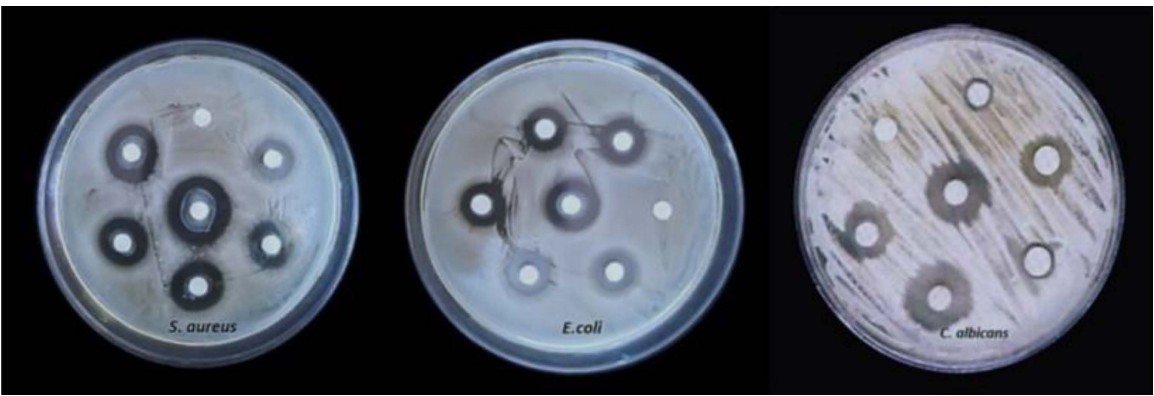

**Fig 8. Sensitivity and minimum inhibitory concentration test (MIC) tests of the synthesized compounds against three clinical isolates organisms.**

Other compounds (including **3e**, **3g**, and **3k**) showed measurable inhibition against one or more organisms, indicating potential antimicrobial activity, whereas compounds **3a**, **3b**, **3d**, **3h**, and **3j** did not produce detectable inhibition zones at the tested concentrations.

**A potential antimicrobial agent against *S. aureus, E. coli*, and *C. albicans*, as determined by resazurin reduction assay method.** The antimicrobial activity of compound 3i was tested against various microorganisms, and MIC values were determined visually using the resazurin dye reduction assay. The change in the dye's colour from blue to pink indicates the viability of microbial cells. The enzyme oxidoreductases present inside unicellular fungi and/or bacterial cells convert resazurin to resorufin, which is pink in colour. When the dye colour remains blue, it indicates that there is no activity in viable cells. The added test material killed bacteria and a fungal cell during incubation, as determined by the blue or purple colour of the dye in the respective wells. The pink colour formation in the wells, even after treatment with the compound 3i and the control drug, indicates the presence of viable cells. Thus, the lowest dilution at which the colour remained blue was taken as the MIC value. The compound was first weighed out and dissolved in dimethyl sulfoxide (DMSO) at a concentration of 50 mg/ml to determine the minimum inhibitory concentration. Compound 3i was tested against each microorganism at 5 μl, and the effect on microbial growth is summarized in Table 8. The results revealed that compound 3i showed better inhibitory activity against microbial growth. The compound 3i was active against *E. coli*, *S. aureus*, and *C. albicans*, as well as against *S. aureus* and *C. albicans*, at concentrations between 0.5–100 μl (0.004–100 μg/ml), as shown in Table 9. The MIC of compound 3i against E. coli (8.35 μg/ml) was lower than that of the antibiotic Streptomycin (10.11 μg/ml). Table 9 presents the minimum inhibitory concentration (MIC) values of compound 3i against three microorganisms: S. aureus, E. coli, and C. albicans, as determined by the resazurin reduction assay. The MIC values represent the lowest concentration of the compound that inhibited the growth of the microorganisms (Fig 9). The MIC values of compound 3i against *S. aureus, E. coli,* and *C. albicans* were 7.13 ± 1.01, 8.35 ± 2.11, and 4.16 ± 1.01 μg/ml, respectively. These values indicate that compound 3i has significant inhibitory activity against all three microorganisms tested. The positive control antibiotics, Streptomycin and Fluconazole, were also tested for comparison. The MIC values of Streptomycin against *S. aureus, E. coli,* and *C. albicans* were found to be 6.91 ± 0.21, 10.11 ± 0.11, and 5.11 ± 3.01 μg/ml, respectively. The MIC values of Fluconazole were not determined for *S. aureus* and *E. coli*, but for *C. albicans*, the MIC was 2.21 ± 0.01 μg/ml. Among the synthesized compounds, compound 3i exhibited the most promising antibacterial activity against both Gram-positive and Gram-negative bacteria at the lowest concentration.

**Table 8. Inhibition zone diameters (mm) of compound 3i against clinical isolates at different concentrations. (Values are mean ± SD, n = 3).**

| Volume of 3i (μl) | *S. aureus* | *E. coli* | *C. albicans* |
|---|---|---|---|
| 100 | 9.41 ± 0.35 | 8.707 ± 0.13 | 8.307 ± 0.13 |
| 50 | 7.12 ± 0.01 | 8.11 ± 1.11 | 8.11 ± 02.21 |
| 25 | 6.81 ± 0.01 | 7.92 ± 0.04 | 7.12 ± 1.20 |
| 12.5 | 5.91 ± 0.01 | 7.01 ± 0.21 | 6.31 ± 1.11 |
| 6.3 | 3.91 ± 0.01 | 6.91 ± 0.10 | 5.91 ± 2.33 |
| 3.1 | 0.63 ± 0.11 | 1.11 ± 0.20 | 2.91 ± 0.11 |
| 1.6 | 0.61 ± 0.01 | 1.201 ± 1.20 | 1.51 ± 0.04 |
| 0.8 | 0.31 ± 0.10 | 0.681 ± 0.10 | 1.721 ± 0.01 |
| 0.4 | 0.051 ± 0.12 | 0.161 ± 1.10 | 1.634 ± 2.11 |
| 0.2 | 0.0051 ± 0.01 | 0.09 ± 0.01 | 1.12 ± 0.11 |
| Strp (5 μl) | 0 | 0 | 0 |
| Flu (5 μl) | 0 | 0 | 0 |
| NC | 100 | 100 | 100 |

**Table 9. Minimum Inhibitory Concentration (MIC) Values of Compound 3i Against *S. aureus, E. coli,* and *C. albicans* Determined by Resazurin Reduction Assay Method.**

| comp | Minimum inhibitory concentration (MIC) by µg/mL | | |
|---|---|---|---|
| | *S. aureus* | *E. coli* | *C. albicans* |
| **3i** | 7.13±1.01 | 8.35±2.11 | 4.16±1.01 |
| P.C (Streptomycin) | 6.91±0.21 | 10.11±0.11 | 5.11±3.01 |
| P.C (Fluconazole) | NZ | NZ | 2.21±0.01 |
| (DMSO) | NZ | NZ | NZ |

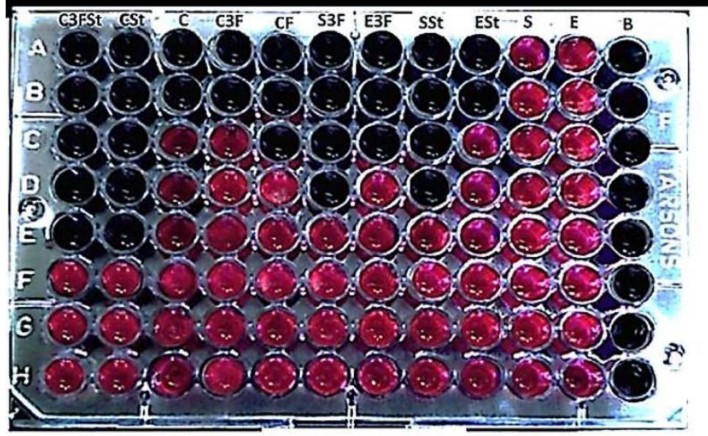

**Fig 9. A digital image that depicts the inhibitory effect of compound 3i on bacterial growth according to the resazurin reduction assay method.** The figure includes a blank control (B) and individual wells containing *E. coli* **(E)**, *S. aureus* **(S)**, and *C. albicans* **(C)**, as well as positive controls of *E. coli* with Streptomycin (ESt) and *S. aureus* with Streptomycin (SSt), and a *C. albicans* with Fluconazole (CF). The image also includes wells containing *E. coli* with compound **3i** (E3f), *S. aureus* with compound **3i** (S3f), *C. albicans* with compound **3i** (C3f), *C. albicans* with compound **3i** and Streptomycin (C3FSt), and *C. albicans* with Streptomycin (CSt).

The higher MIC observed in the resazurin assay suggests that compound **3i** may have reduced solubility or a tendency to aggregate in the aqueous broth environment compared to its diffusion in agar, or that the sensitive resazurin endpoint detected partial growth inhibition not observable as a clear zone on agar. Therefore, while the agar well diffusion method was valuable for initial screening, the values obtained from the resazurin broth microdilution assay represent the true, definitive MICs for compound **3i**.

The blue or purple colour of the dye in the wells indicates bacterial or fungal growth inhibition, while the pink colour indicates the presence of viable cells. Fig 9 shows that compound 3i exhibits inhibitory activity against all three microorganisms tested. *E. coli* and *S. aureus* showed a significant reduction in growth in the presence of the compound. The figure also demonstrates that compound 3i has an inhibitory effect on C. albicans similar to that of Fluconazole, a commonly used antifungal drug.

## Statistical analysis

All antimicrobial assays, including inhibition zone measurements and MIC determinations, were performed in triplicate (n = 3). Each replicate was conducted using an independently prepared agar plate to account for plate-to-plate variation. Results are presented as mean±standard deviation (SD). Statistical analysis was performed using one-way analysis of

variance (ANOVA) to evaluate differences among groups. When significant differences were detected, Tukey's HSD post hoc test was applied for pairwise comparisons. A value of $p < 0.05$ was considered statistically significant.

### The anti-biofilm activity of compound 3i against *S. aureus, E. coli,* and *C. albicans* biofilms

The anti-biofilm activity of compound 3i was evaluated against *S. aureus*, *E. coli*, and *C. albicans* using the crystal violet assay. To distinguish specific anti-biofilm effects from general growth inhibition, compound **3i** was tested at sub-MIC concentrations ($0.5 \times$ MIC) that do not affect planktonic cell viability. Biofilms were formed by inoculating the isolates into biofilm media in the presence or absence of compound 3i, and biofilm biomass was quantified by measuring optical density (OD) at 570 nm. All experiments were performed in triplicate across three independent experiments ($n = 3$ per condition), and results are presented as mean ± standard deviation (SD).

As shown in Fig 10 and Table 10, compound **3i** significantly reduced biofilm formation in all three microorganisms compared to untreated controls. For *S. aureus*, biofilm $OD_{570}$ decreased from 0.42 ± 0.04 (Untreated) to 0.19 ± 0.03 (treated), corresponding to 54.8% inhibition ($p \leq 0.001$). For *E. coli*, $OD_{570}$ decreased from 0.61 ± 0.05 to 0.35 ± 0.04 (42.6% inhibition; $p \leq 0.001$). For *C. albicans*, $OD_{570}$ decreased from 0.28 ± 0.03 to 0.12 ± 0.02 (57.1% inhibition; $p \leq 0.001$). Representative images in Fig 10 illustrate the visible reduction in biofilm density following treatment.

As compound **3i** was tested at sub-MIC concentrations that do not inhibit planktonic growth, the observed reduction in biofilm biomass suggests specific anti-biofilm activity rather than a secondary consequence of inhibition of bacterial or fungal growth.

### Conclusion

In summary, a series of biaryl (biphenyl) analogs was synthesized via Suzuki coupling and evaluated for *in vitro* antimicrobial activity against *Staphylococcus aureus, Escherichia coli*, and *Candida albicans*, as well as for biofilm inhibition.

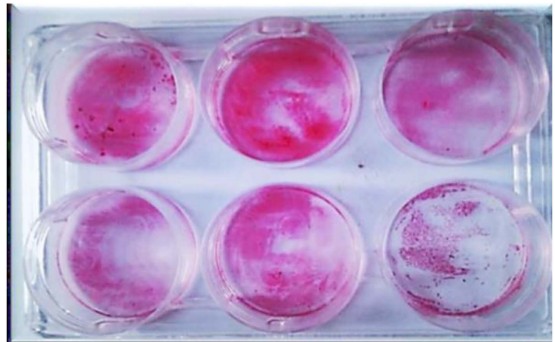

**Fig 10. The Effect of compound 3i on biofilm formation density of clinical isolates of *S. aureus*, *E. coli*, and *C. albicans*.**

**Table 10. Optical Density (OD) Values of Biofilms of *S. aureus, E. coli,* and *C. albicans* before and after treatment with compound 3i.**

| Isolates microbes | OD values | |
|---|---|---|
| | Untreated-3i | Treated-3i |
| *S. aureus* | 0.42 ± 0.04 | 0.19 ± 0.03 |
| *E. coli* | 0.61 ± 0.05 | 0.35 ± 0.04 |
| *C. albicans* | 0.28 ± 0.03 | 0.12 ± 0.02 |

Among the tested compounds, **3i** showed the most consistent activity in the antibacterial assays and measurable effects in the anti-biofilm evaluation, supported by the *in silico* interaction analysis. These findings identify compound **3i** as a promising lead scaffold for further structure optimization and expanded biological evaluation, including standardized susceptibility testing across broader and well-characterized isolate panels and assessment of cytotoxicity/selectivity before any therapeutic claims can be made.

## Supporting information

**S1 File. Supplemental 1S, 2S, 3S, 4S, 5S, 6S and 7S figures.**
(DOCX)

## Acknowledgments

The authors extend their appreciation to the Deputyship for Research & Innovation, Ministry of Education in Saudi Arabia, for supporting this research work through the project number: IFP22UQU4331005DSR02.

## Author contributions

**Data curation:** Yasser Mohammed.

**Methodology:** Ahmed Hassen Shntaif.

**Project administration:** Yasser Mohammed.

**Writing – original draft:** Saad Alghamdi.

**Writing – review & editing:** Ahmed Hassen Shntaif.

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
