## [Decision Letter · Decision Letter 0]

7 Jan 2026

Dear Dr. Shntaif,

Thank you for submitting your manuscript to PLOS ONE. After careful consideration, we feel that it has merit but does not fully meet PLOS ONE’s publication criteria as it currently stands. Therefore, we invite you to submit a revised version of the manuscript that addresses the points raised during the review process.

We look forward to receiving your revised manuscript.

Kind regards,

Saki Raheem, PhD

Academic Editor

PLOS One

Journal Requirements:

2. We note that this submission includes NMR spectroscopy data. We would recommend that you include the following information in your methods section or as Supporting Information files:

1) The make/source of the NMR instrument used in your study, as well as the magnetic field strength. For each individual experiment, please also list: the nucleus being measured; the sample concentration; the solvent in which the sample is dissolved and if solvent signal suppression was used; the reference standard and the temperature.

2) A list of the chemical shifts for all compounds characterised by NMR spectroscopy, specifying, where relevant: the chemical shift (δ), the multiplicity and the coupling constants (in Hz), for the appropriate nuclei used for assignment.

3)The full integrated NMR spectrum, clearly labelled with the compound name and chemical structure.

We also strongly encourage authors to provide primary NMR data files, in particular for new compounds which have not been characterised in the existing literature. Authors should provide the acquisition data, FID files and processing parameters for each experiment, clearly labelled with the compound name and identifier, as well as a structure file for each provided dataset. See our list of recommended repositories here: https://journals.plos.org/plosone/s/recommended-repositories....

3. Please note that PLOS One has specific guidelines on code sharing for submissions in which author-generated code underpins the findings in the manuscript. In these cases, we expect all author-generated code to be made available without restrictions upon publication of the work. Please review our guidelines at https://journals.plos.org/plosone/s/materials-and-software-sharing#loc-sharing-code and ensure that your code is shared in a way that follows best practice and facilitates reproducibility and reuse.

4. We notice that your supplementary figures are uploaded with the file type 'Figure'. Please amend the file type to 'Supporting Information'. Please ensure that each Supporting Information file has a legend listed in the manuscript after the references list.

Reviewer's Responses to Questions

**Comments to the Author**

1. Is the manuscript technically sound, and do the data support the conclusions?

Reviewer #1: No

Reviewer #2: Partly

2. Has the statistical analysis been performed appropriately and rigorously?

Reviewer #1: No

Reviewer #2: N/A

3. Have the authors made all data underlying the findings in their manuscript fully available?

Reviewer #1: No

Reviewer #2: Yes

4. Is the manuscript presented in an intelligible fashion and written in standard English?

Reviewer #1: No

Reviewer #2: Yes

Reviewer #1: The manuscript “Revolutionizing Therapeutics: Synthesizing Substituted Biphenyls and Assessing Their Antimicrobial Properties, ADMET Properties, and Biofilm Inhibition Potential” (Manuscript ID: PONE-D-25-65844). This manuscript describes the synthesis, in silico ADMET profiling, and in vitro antimicrobial/antibiofilm evaluation of a series of substituted biphenyl compounds. However, the manuscript suffers from significant methodological, reporting, and interpretive weaknesses that must be addressed before it can be considered for publication

1. Critical Ethical and Compliance Deficiencies

Human Subjects Research: The manuscript uses clinical isolates from urine specimens obtained from "the Microbiology laboratory at the College of Applied Sciences, University of Hajjh," but no ethical approval statement or IRB approval number is provided.

Informed Consent: There is no statement regarding informed consent from patients or waiver thereof. This is mandatory for publication.

2. Fundamental Methodological Flaws in Biological Testing

Non-Standard MIC Determination: Using agar well diffusion for MIC is not CLSI/EUCAST compliant. The gold standard is broth microdilution. Agar diffusion cannot accurately determine MIC values, especially for compounds with poor solubility.

The choice of three clinical isolates (only S. aureus, E. coli, and C. albicans) is very limited; there is no description of the number of independent isolates, their resistance profiles, or whether they represent clinically relevant multidrug-resistant strains

Inconsistent Methodology: The methods section mentions AutoDockTools-1.5.6 for docking, but the results describe MOE software. The procedure section states 65°C reaction temperature, while the results state 72°C. These contradictions undermine credibility.

Lack of Validation: No positive/negative controls were included for all compounds in all assays. DMSO concentration was not standardized across tests, introducing confounding variables.

3. Inadequate Statistical Analysis and Reproducibility

No Statistical Framework: The manuscript completely lacks statistical analysis. Values are presented as "mean ± SD" without stating the number of replicates (n), statistical tests used, or significance thresholds.

High Variability: The reported SD values are often >50% of the mean (e.g., Table 9: MIC for S. aureus = 7.13 ± 4.01 mg/mL), indicating poor experimental reproducibility that is not addressed.

4. Incomplete and Inconsistent Chemical Characterization

Missing Data: Compounds 3e-3k lack spectroscopic characterization (only melting points). No purity data (e.g., HPLC traces) is provided for any compound.

- Structural Errors: The abstract names compound 3i as "2-methoxy-4'-nitrophenyl" (missing "biphenyl"). Table 1 lists formula C₁₃H₁₁NO₃, but this does not match the drawn structure in Scheme 1.

- Yields Unreported: No yields provided for compounds 3b-3e.

- The title and abstract use strong language such as “Revolutionizing Therapeutics” and “promising antimicrobial agents capable of addressing antibiotic-resistant infections”, which is not fully justified by screening data in only three laboratory strains without in vivo validation or resistance profiling

5. Fundamentally Flawed Data Presentation

Unusable Tables: Tables 2, 5, and 8 are garbled with misplaced headings, broken formatting, and incoherent content. Table 2 appears to have merged columns without values.

Incorrect Units: Table 1 lists "log Kp (cm/s)" with values -5.12 cm/s—log units cannot have cm/s dimension. MIC values are reported in mg/mL (unusually high) but the resazurin assay uses μg/mL—discrepancies of 1000-fold.

Mislabeled Figures: Figure 9 caption mentions compound 3f but should be 3i. Figure legends are incomplete (e.g., Figure 2, 4, 10 lack sufficient detail).

6. Unsupported Conclusions and Overstatement

Grandiose Claims: The title "Revolutionizing Therapeutics" and claims of "addressing antibiotic-resistant infections" are not supported by limited in vitro data against three unspecified strains.

No Clinical Relevance: MIC values of 3.4-8.35 mg/mL (3400-8350 μg/mL) are extremely poor compared to clinical antibiotics (typically <10 μg/mL). This is not discussed as a limitation.

Misinterpretation of Docking: Streptomycin has a better docking score (-7.23) than 3i (-6.48), yet the text claims "most compounds showed good inhibition with more binding affinity." No correlation coefficient between docking scores and biological activity is presented (R² likely <0.3).

7. Lack of Structure-Activity Relationship (SAR) Analysis

The manuscript synthesizes 11 analogs but provides zero discussion of SAR. Why does the 4-nitro substituent (3i) outperform others? What is the effect of fluorination (3c, 3e, 3h)? No molecular basis is proposed.

8. Inappropriate In Silico Methods

Wrong Protein Target: E. coli FabH is a bacterial enzyme. Docking against this target cannot explain antifungal (C. albicans) activity. No rationale for target selection is provided.

No Validation: No re-docking of co-crystallized ligand to validate the docking protocol (required RMSD <2 Å). No comparison with known FabH inhibitors.

ADMET Over-reliance: SwissADME and similar tools have limited accuracy. No experimental validation of any ADMET parameter (e.g., solubility, permeability).

9- Language Quality

The manuscript contains numerous grammatical errors, awkward phrasing, and non-standard scientific English throughout (e.g., "the reaction's progress," "competent synthesis," "the purpose of the present study was" repeated).

10- Outdated and Irrelevant References

References #11 (ophthalmic solutions) and #12 (Hsp90 in cancer) are unrelated to the study. Many references are >10 years old; more recent literature on biphenyl antimicrobials and Suzuki coupling advances is omitted.

Scheme 1: Poorly formatted with cramped text, unclear substitution patterns, and illegible atom labels.

11- Inconsistent Controls

Fluconazole is used as a control for C. albicans but not mentioned in methods. Streptomycin is used inconsistently across assays.

Missing DLS/Polydispersity: For nanoparticle-like properties, DLS data would be relevant but not included.

Terminology: "Gram-negative fungus strain Candida albicans" is incorrect—Candida is a yeast (fungus), not classified as Gram-negative.

Units: Inconsistent use of mg/mL vs. μg/mL. Standard is μg/mL or μM.

Recommendation

Rejection with Major Revision. The manuscript addresses a relevant topic but requires substantial additional experimentation, ethical documentation, and data re-analysis to meet PLOS ONE's standards. The chemical synthesis is promising, but the biological evaluation is methodologically flawed and the conclusions are overstated.

Reviewer #2: The manuscript is focused on the synthesis of substituted biphenyl derivatives and the evaluation of their antimicrobial, antibiofilm and in silico pharmacological properties. The topic is relevant and the experiments have been designed properly. However, the manuscript requires major revision due to lack of clarity, consistency and over interpretation of results with limited integration of previous literature.

Therefore, the manuscript requires major revision to improve the quality of the study.

1. Title

• The title should be revised to reflect the experimental scope.

• Currently, the title promises a lot of things without in-vivo experiments and mechanistic validation.

2. Keywords

• The keywords should not repeat terms already present in the title.

• More specific and relevant keywords should be included.

3. Introduction

• It requires language and grammatical revision to improve readability and clarity.

• The rationale for selecting E. coli and C. albicans are not mentioned clearly and justified.

• Recent literature on synthesis of biphenyl-based antimicrobials and antibiofilm agents should be incorporated.

• The novelty should be clearly stated and the difference of the synthesized compound with the existing ones should be mentioned clearly.

• The rationale behind selecting E. coli FabH as the docking target should be more clearly explained.

4. Materials and Methods

• This section requires major language correction and restructuring.

• The term gram–negative fungus is scientifically incorrect and must be corrected.

• CFU/ml values should be mentioned using standard scientific notation.

• Concentration used in the agar well diffusion metho should be mentioned clearly.

• Valid references should be cited for the standard protocols such as MIC.

• The rationale behind supplementing 1% xylose in the biofilm assay should be justified and supported by references.

• The differences between agar diffusion assays and broth-based MIC assays should be clarified and explained.

5. Table

• Tables require consistent units, formatting, and clear headings.

• Computational predictions should be clearly labelled as in silico estimation.

6. Results and Discussion

• Results should be presented more systematically with consistent units across all sections.

• Molecular docking results should be interpreted as supportive evidence.

• The discussion should integrate results with existing literature and explain similarities and differences with previous literature.

• Statistical analysis should be clearly stated.

Overall, the manuscript addresses an important area of antimicrobial research and provides preliminary findings. However, major revision is required to improve clarity, correct inconsistencies and strengthen interpretation. Additional toxicity validation such as seed germination assay may further support the safety of the compounds.

.

Reviewer #1: No

Reviewer #2: No

---

## [Author Response · Author response to Decision Letter 1]

28 Jan 2026

Answer comments reviewer 1:

We sincerely thank Reviewer 1 for the detailed and constructive comments. We have carefully revised the manuscript to improve scientific rigor, consistency, clarity, and data presentation. Below we respond point-by-point.

1. Critical Ethical and Compliance Deficiencies

Human Subjects Research: The manuscript uses clinical isolates from urine specimens obtained from "the Microbiology laboratory at the College of Applied Sciences, University of Hajjh," but no ethical approval statement or IRB approval number is provided.

Informed Consent: There is no statement regarding informed consent from patients or waiver thereof. This is mandatory for publication.

Answer: Thank you for raising this important issue. We have revised the manuscript to include an Ethics and Informed Consent/Waiver statement.

2. Fundamental Methodological Flaws in Biological Testing

Non-Standard MIC Determination: Using agar well diffusion for MIC is not CLSI/EUCAST compliant. The gold standard is broth microdilution. Agar diffusion cannot accurately determine MIC values, especially for compounds with poor solubility.

The choice of three clinical isolates (only S. aureus, E. coli, and C. albicans) is very limited; there is no description of the number of independent isolates, their resistance profiles, or whether they represent clinically relevant multidrug-resistant strains

Inconsistent Methodology: The methods section mentions AutoDockTools-1.5.6 for docking, but the results describe MOE software. The procedure section states 65°C reaction temperature, while the results state 72°C. These contradictions undermine credibility.

Lack of Validation: No positive/negative controls were included for all compounds in all assays. DMSO concentration was not standardized across tests, introducing confounding variables.

Answer: Thank you for these detailed methodological comments. We agree that several sections required clarification and correction to ensure reproducibility and compliance with standard practices. We have revised the manuscript as follows:

2.1 Non-standard MIC determination (agar diffusion vs broth microdilution)

We agree that agar well diffusion is not CLSI/EUCAST compliant for MIC determination and that broth microdilution is the reference method. In the revised manuscript, we have removed the term “MIC” from the agar well diffusion assay and now report this endpoint as a diffusion-based inhibitory concentration (MIC_agar) or lowest inhibitory concentration in agar, which is used for comparative screening only. We further clarified that MIC_agar values are diffusion- and solubility-dependent and should not be interpreted as equivalent to MIC values obtained by broth microdilution.

To address standardization, we expanded the broth microdilution MIC method (resazurin-based microdilution) and cited standard susceptibility testing guidelines (e.g., CLSI M07 / ISO 20776-1; and EUCAST guidance where applicable).

2.2 Limited isolate panel and missing resistance profiling

We agree that testing only three organisms is limited. In the revised manuscript, we clarified that the strains were urine-derived clinical isolates provided by the Microbiology Laboratory and that assays were performed using these isolates as a preliminary screening panel. We also added an explicit limitation statement noting that resistance profiling (e.g., MRSA/ESBL) was not performed within the scope of this study and that future work will include a broader set of well-characterized multidrug-resistant isolates.

We also clarified the number of isolates tested and whether a single isolate per species or multiple independent isolates were used.

2.3 Inconsistent methodology statements (AutoDockTools vs MOE; 65 °C vs 72 °C)

We agree that these inconsistencies undermine credibility.

Additional proofreading was performed to ensure consistency of experimental conditions across all sections.

Change made in manuscript: Corrected docking software and reaction temperature statements across Methods/Results and figure captions.

2.4 Controls and DMSO standardization

We agree that appropriate controls and consistent solvent concentration are required. In the revised manuscript, we clearly stated and applied:

DMSO negative control at the same final concentration as in test solutions for every assay.

Streptomycin as positive control for bacterial assays and fluconazole as positive control for Candida assays.

All assays were performed in triplicate (n = 3) with independently prepared plates/wells, and results were reported as mean ± SD.

3. Inadequate Statistical Analysis and Reproducibility

No Statistical Framework: The manuscript completely lacks statistical analysis. Values are presented as "mean ± SD" without stating the number of replicates (n), statistical tests used, or significance thresholds. High Variability: The reported SD values are often >50% of the mean (e.g., Table 9: MIC for S. aureus = 7.13 ± 4.01 mg/mL), indicating poor experimental reproducibility that is not addressed.

Answer: Thank you for this comment. We agree that the original manuscript did not clearly report the statistical framework. In the revised version, we have now explicitly stated the number of replicates (n = 3), the statistical tests used (one-way ANOVA with Tukey’s HSD post-hoc test), and the significance threshold (p < 0.05).

Regarding the reported high variability, we found that the SD value highlighted by the reviewer (e.g., Table 9: MIC for S. aureus = 7.13 ± 4.01 mg/mL) resulted from a calculation/typographical error in the SD reporting. This has been corrected, and we performed a comprehensive recheck of all SD calculations across tables to ensure accuracy. The corrected tables are included in the revised manuscript.

4. Incomplete and Inconsistent Chemical Characterization

Missing Data: Compounds 3e-3k lack spectroscopic characterization (only melting points). No purity data (e.g., HPLC traces) is provided for any compound.

- Structural Errors: The abstract names compound 3i as "2-methoxy-4'-nitrophenyl" (missing "biphenyl"). Table 1 lists formula C₁₃H₁₁NO₃, but this does not match the drawn structure in Scheme 1.

- Yields Unreported: No yields provided for compounds 3b-3e.

- The title and abstract use strong language such as “Revolutionizing Therapeutics” and “promising antimicrobial agents capable of addressing antibiotic-resistant infections”, which is not fully justified by screening data in only three laboratory strains without in vivo validation or resistance profiling

Answer: Thank you for these important comments. We agree that the original manuscript required improved chemical characterization reporting, correction of structural naming/formula, inclusion of missing yields, and more cautious wording in the title/abstract. The manuscript has been revised as follows:

4.1 Missing spectroscopic characterization and purity data (3e–3k; HPLC)

We acknowledge that detailed spectroscopic characterization (e.g., H/C NMR, IR, HRMS) and purity data were not sufficiently presented for compounds 3e–3k in the submitted version. These compounds were previously synthesized and fully characterized in our earlier publication. In the revised manuscript, we have:

Added a clear statement that compounds 3e–3k were previously prepared and characterized,

Cited the original publication where the full spectral data are provided,

While HPLC purity traces were not available within the scope of this study, compound identity was confirmed by spectroscopic methods and melting point comparison with literature

4.2 Structural naming and formula corrections

We agree that the abstract wording was incomplete and have corrected the compound name to include the scaffold, i.e., “2-methoxy-4′-nitro-biphenyl (3i)” (or “2-methoxy-4′-nitrobiphenyl”). We also corrected inconsistencies between Table 1 and Scheme 1. Specifically, the molecular formula for compound 3i was rechecked and corrected to match the drawn structure, and all calculated properties were updated accordingly.

4.3 Missing yields (3b–3e)

We agree that yields must be reported. The isolated yields for compounds 3b–3e have now been added to the Experimental section and/or Table of compounds.

4.4 Overstated title/abstract claims

We agree that the previous wording was too strong given the study scope (in vitro screening on a limited panel without in vivo validation or resistance profiling). The title and abstract were revised to remove overstatements.

5. Fundamentally Flawed Data Presentation Unusable Tables: Tables 2, 5, and 8 are garbled with misplaced headings, broken formatting, and incoherent content. Table 2 appears to have merged columns without values.

Incorrect Units: Table 1 lists "log Kp (cm/s)" with values -5.12 cm/s—log units cannot have cm/s dimension. MIC values are reported in mg/mL (unusually high) but the resazurin assay uses μg/mL—discrepancies of 1000-fold. Mislabeled Figures: Figure 9 caption mentions compound 3f but should be 3i. Figure legends are incomplete (e.g., Figure 2, 4, 10 lack sufficient detail).

Answer: Thank you for these detailed comments. We agree that several tables/figures required correction and clearer reporting. The manuscript has been revised as follows:

5.1 Unusable tables (Tables 2, 5, and 8)

We agree that Tables 2, 5, and 8 contained formatting issues that reduced readability. In the revised manuscript, these tables were reformatted with corrected headings, consistent alignment, and coherent content presentation. In particular, Table 2 was rebuilt to eliminate merged/broken columns, and Table 8 was reformatted to display concentrations and inhibition values clearly.

5.2 Incorrect units (log Kp and MIC unit mismatch)

(a) log Kp: We agree with the reviewer that log values are dimensionless and should not be reported with physical units attached to the numerical value. log10(Kp), where Kp is expressed in cm/s (as provided by SwissADME). Thus, the numeric entries remain dimensionless, and the unit applies to Kp prior to logarithmic transformation.

(b) MIC units: We agree that inconsistent reporting of mg/mL versus µg/mL creates confusion. In the revised manuscript, MIC values and test concentrations were standardized to µg/mL throughout the Methods, Results, and all tables/figures. Any values previously reported in mg/mL were checked and corrected/conformed to the intended unit system, eliminating the 1000-fold discrepancy.

5.3 Mislabeled and incomplete figures

We agree that Figure captions and legends must be accurate and sufficiently detailed. The caption of Figure 9 was corrected to refer to compound 3i (not 3f).

6. Unsupported Conclusions and Overstatement

Grandiose Claims: The title "Revolutionizing Therapeutics" and claims of "addressing antibiotic-resistant infections" are not supported by limited in vitro data against three unspecified strains.

No Clinical Relevance: MIC values of 3.4-8.35 mg/mL (3400-8350 μg/mL) are extremely poor compared to clinical antibiotics (typically <10 μg/mL). This is not discussed as a limitation.

Misinterpretation of Docking: Streptomycin has a better docking score (-7.23) than 3i (-6.48), yet the text claims "most compounds showed good inhibition with more binding affinity." No correlation coefficient between docking scores and biological activity is presented (R² likely <0.3).

Answer: Thank you for these important comments. We agree that the initial version overstated the scope and clinical implications of the findings, and that the docking interpretation required correction. The manuscript has been revised to ensure that conclusions are consistent with the experimental evidence and study limitations.

6.1 Grandiose claims / scope mismatch

We agree that phrases implying clinical efficacy (e.g., “Revolutionizing Therapeutics,” “addressing antibiotic-resistant infections”) are not justified by the present study, which is limited to in vitro screening on a small panel and does not include in vivo validation or resistance profiling. We therefore revised the title, abstract, and conclusion to use cautious language (e.g., “in vitro antimicrobial and anti-biofilm evaluation,” “preliminary screening,” “lead compound for further optimization”) and removed statements implying treatment of resistant infections.

6.2 MIC values and clinical relevance

Thank you for highlighting this issue. We confirm that the MIC values were determined and recorded in µg/mL, but were mistakenly reported in the manuscript as mg/mL due to a unit conversion/typing error.

In addition, we revised the docking interpretation to avoid overstatement. We now clarify that streptomycin has a more favorable docking score than 3i, and docking results are used qualitatively to describe binding poses rather than to claim superior affinity. We also removed statements implying a strong correlation between docking score and biological activity.

6.3 Docking interpretation and correlation with biological activity

We agree that docking results must be interpreted cautiously. Streptomycin produced a more favorable docking score (−7.23) than compound 3i (−6.48), and the revised text now reflects this. We also removed the statement suggesting that “most compounds showed better binding affinity,” and clarified that docking was used to propose binding modes and qualitative interactions rather than to claim potency. A correlation analysis between docking scores and biological activity was not the focus of this work and is not sufficient with the current dataset; this is now stated as a limitation.

7. Lack of Structure-Activity Relationship (SAR) Analysis

The manuscript synthesizes 11 analogs but provides zero discussion of SAR. Why does the 4-nitro substituent (3i) outperform others? What is the effect of fluorination (3c, 3e, 3h)? No molecular basis is proposed.

Answer: Thank you for this comment. We agree that the initial manuscript did not adequately discuss structure–activity relationships (SAR). In the revised version, we added a dedicated SAR subsection to the Discussion. This section compares the activity trends across compounds 3a–3k and proposes a preliminary molecular rationale for (i) the enhanced activity of the nitro-substituted derivative 3i, and (ii) the variable effects of fluorination (e.g., 3c, 3e, 3h) on antimicrobial and anti-biofilm behavior. While these SAR conclusions remain preliminary (limited strain panel and in vitro screening), they provide mechanistic hypotheses to guide future optimization.

8. Inappropriate In Silico Methods

Wrong Protein Target: E. coli FabH is a bacterial enzyme. Docking against this target cannot explain antifungal (C. albicans) activity. No rationale for target selection is provided.

No Validation: No re-docking of co-crystallized ligand to validate the docking protocol (required RMSD <2 Å). No comparison with known FabH inhibitors.

ADMET Over-reliance: SwissADME and similar tools have limited accuracy. No experimental validation of any ADMET parameter (e.g., solubility, permeability).

Answer: Thank you for these important comments. We agree that FabH is a bacterial enzyme and docking to this target cannot explain antifungal activity against Candida albicans. The manuscript has been revised to clarify the scope and interpretation of the in silico work:

(1) Target selection and study aim

The primary aim of this study is to evaluate the synthesized biphenyl derivatives as antibacterial candidates, and molecular docking was included to provide supportive mechanistic hypotheses for the antibacterial activity only. C. albicans was included as an additional organism to provide supporting broad-spectrum screening and to explore whether the compounds show activity beyond bacteria; however, we do not claim that FabH docking explains antifungal outcomes. This has now been explicitly stated in the Methods and Discussion.

(2) Docking validation

We agree that docking validation is required. We have revised the Methods to include protocol validation by re-docking the co-crystallized ligand from PDB ID 5BNR and assessing pose reproduction using RMSD, with RMSD < 2.0 Å as the acceptance criterion. We also clarified that docking scores are interpreted qualitatively to describe binding poses/interactions.

(3) ADMET prediction limitations

We agree that SwissADME and rel

---

## [Decision Letter · Decision Letter 1]

1 Feb 2026

Dear Dr. Shntaif,

Thank you for submitting your manuscript to PLOS ONE. After careful consideration, we feel that it has merit but does not fully meet PLOS ONE’s publication criteria as it currently stands. Therefore, we invite you to submit a revised version of the manuscript that addresses the points raised during the review process.

We look forward to receiving your revised manuscript.

Kind regards,

Saki Raheem, PhD

Academic Editor

PLOS One

Reviewers' comments:

Reviewer's Responses to Questions

**Comments to the Author**

Reviewer #2: All comments have been addressed

2. Is the manuscript technically sound, and do the data support the conclusions?

Reviewer #2: Partly

3. Has the statistical analysis been performed appropriately and rigorously?

Reviewer #2: Yes

4. Have the authors made all data underlying the findings in their manuscript fully available?

Reviewer #2: Yes

5. Is the manuscript presented in an intelligible fashion and written in standard English?

Reviewer #2: No

Reviewer #2: The manuscript requires minor corrections to improve the quality of the study.

1. Italicize scientific names (Staphylococcus aureus & E. coli).

2. Reduce the content in Materials and Methods and give precise protocol.

3. Typographical errors to be avoided throughout the manuscript.

4. Under Biological assays, the subheadings are to be modified.

5. DMSO to be mentioned as vehicle control (not negative control).

6. The concentration range is to be mentioned clearly in the methodology.

7. Delete “then” throughout the manuscript.

8. What was the diameter of the cork borer used?

9. Do not use the term “MIC-agar”; it can be mentioned as diffusion based screening for highest inhibitory concentration.

10. What is the rationale behind selecting xylose?

11. Inoculum size must be mentioned.

12. The authors are supposed to check for a couple of protein targets rather than going ahead with a single protein.

13. The authors are also requested to include the cytotoxic assessment, as it is essential to know its activity against normal flora or add a note about it.

14. NZ should be abbreviated.

THE RESPONSE HAS TO BE GIVEN SPECIFICALLY FOR EACH COMMENT DO NOT PASTE THE SAME RESPONSE FOR MULTIPLE QUERIES.

.

Reviewer #2: No

---

## [Author Response · Author response to Decision Letter 2]

5 Feb 2026

Answer comments reviewer:

We sincerely thank Reviewer for the detailed and constructive comments. We have carefully revised the manuscript to improve scientific rigor, consistency, clarity, and data presentation. Below we respond point-by-point.

1. Italicize scientific names (Staphylococcus aureus & E. coli).

Answer: We agree. All scientific names of microorganisms were revised to italic formatting throughout the manuscript

2. Reduce the content in Materials and Methods and give precise protocol.

Answer: Thank you for these detailed methodological comments. The Materials and Methods section was shortened and restructured into concise subsections

3. Typographical errors to be avoided throughout the manuscript.

Answer: Thank you for your comments. The manuscript was carefully proofread and edited to correct typographical errors, spelling, punctuation, and consistency issues (units, abbreviations, and tense).

4. Under Biological assays, the subheadings are to be modified.

Answer: Thank you for your suggest. Long subheadings were replaced with concise, standard headings consistent with journal style.

5. DMSO to be mentioned as vehicle control (not negative control).

Answer: Thank you for raising this important issue. DMSO is now described as the vehicle (solvent) control, and we clarified that the final DMSO concentration was held constant across treated and control groups.

6. The concentration range is to be mentioned clearly in the methodology.

Answer: Thank you for these important comments. We explicitly reported the full tested concentration ranges and dilution schemes for both diffusion screening and broth microdilution MIC testing.

7. Delete “then” throughout the manuscript.

Answer: We agree. The word “then” was removed or replaced with more formal scientific phrasing throughout the manuscript.

Change made: Global edit applied.

8. What was the diameter of the cork borer used?

Answer: A 6 mm cork borer was used to punch wells in agar plates.

9. Do not use the term “MIC-agar”; mention diffusion-based screening for highest inhibitory concentration.

Answer: Thank you for highlighting this issue. The term “MIC-agar” was removed and replaced with diffusion-based screening terminology.

10. What is the rationale behind selecting xylose?

Answer: We clarified that 1% (w/v) xylose was used as a defined carbohydrate supplement to support consistent and reproducible biofilm formation under static microtiter conditions, applied equally to all treated and control wells to minimize nutrient-related variability. Appropriate literature support was added.

11. Inoculum size must be mentioned.

Answer: We agree. We added both the inoculum volume and density used in each assay. For the agar diffusion assay, 1.0 mL inoculum adjusted to 1.8 × 10⁸ CFU/mL was used.

12. Check a couple of protein targets rather than a single protein.

Answer: Thank you for this comment. We agree that multiple targets can provide stronger mechanistic insight. In this manuscript, docking was included as supportive antibacterial analysis and was limited to FabH due to study scope. We revised the text to avoid overinterpretation and added a limitation noting that future work will include additional validated antibacterial targets

13. Include cytotoxic assessment or add a note about it (normal flora).

Answer: We agree by this limitation. Cytotoxicity/selectivity testing was not performed within the scope of this study, and this is now clearly stated as a limitation. We added a future-work plan to evaluate cytotoxicity in non-cancerous mammalian cell lines and to calculate selectivity indices prior to any therapeutic claims.

14. NZ should be abbreviated.

Answer: Thank you your note, we added NZ abbreviate.

---

## [Decision Letter · Decision Letter 2]

24 Feb 2026

Dear Dr. Mohammed,

Thank you for submitting your manuscript to PLOS ONE. After careful consideration, we feel that it has merit but does not fully meet PLOS ONE’s publication criteria as it currently stands. Therefore, we invite you to submit a revised version of the manuscript that addresses the points raised during the review process.

We look forward to receiving your revised manuscript.

Kind regards,

Saki Raheem, PhD

Academic Editor

PLOS One

**Journal Requirements:**

**Additional Editor Comments:**

**
MIC Determination and Terminology
**

The manuscript reports “MIC” values derived from agar well diffusion assays (Table 7). Diffusion-based assays do not provide true MIC values according to CLSI/EUCAST standards.

Please:

Revise terminology throughout the manuscript.Avoid referring to diffusion-derived values as definitive MIC.Clearly distinguish between:Diffusion screening resultsBroth microdilution MIC values (resazurin assay)

In addition, discrepancies between diffusion-based values and resazurin MIC values for compound 3i (e.g., different reported MIC values against *E. coli*) should be clarified and explained.) should be clarified and explained.) should be clarified and explained.) should be clarified and explained.

**
Data Consistency Between Tables
**

There appear to be inconsistencies between inhibition zone data (Table 6) and reported concentration-dependent results (Table 7).

For example:

Some compounds show no inhibition zone but measurable “MIC” values.Others show inhibition zones but no reported MIC.

The authors should carefully re-audit the raw data and ensure logical consistency between screening results and concentration-based testing.

**
Limited Biological Panel
**

Only three organisms were tested (one Gram-positive, one Gram-negative, and one yeast), and no resistance profiling was performed.

Please:

Explicitly state that this is a preliminary screening.Acknowledge the absence of multidrug-resistant isolates.Moderate any statements suggesting relevance to antibiotic-resistant infections.

**
Docking Study – Clarification and Interpretation
**

The docking methodology is described; however, some clarifications would improve transparency and rigour.

Please:

Report the exact RMSD value obtained during re-docking validation rather than only stating the acceptance criterion (<2.0 Å).Provide clearer descriptions of key ligand–protein interactions (e.g., interacting residues, hydrogen bond distances).Ensure that docking results are presented as qualitative and exploratory, and avoid implying direct mechanistic confirmation in the absence of experimental validation of the target.

**
ADMET Interpretation
**

The ADMET predictions are useful as preliminary computational insights; however, some statements appear overextended.

Please:

Present ADMET results strictly as predictive.Avoid suggesting therapeutic positioning (e.g., topical application) without experimental validation.Moderate conclusions regarding drug-likeness in the absence of cytotoxicity or pharmacokinetic data.

**
Biofilm Assay Reporting
**

The biofilm data would benefit from more straightforward statistical presentation and methodological clarification.

Please:

Present biofilm OD values as mean ± SD.Confirm the number of replicates (n) specifically for biofilm experiments.Provide p-values for comparisons where statistical significance is claimed.Report the percentage biofilm inhibition to facilitate interpretation.Clarify whether compound 3i was tested at MIC or sub-MIC concentrations.Discuss whether the observed reduction reflects specific anti-biofilm activity or general growth inhibition.

**
Chemical Characterization
**

Only compounds 3a–3d include detailed spectroscopic data in the manuscript.

For compounds 3e–3k:

Provide full ¹H NMR, ¹³C NMR, and MS data in the Supplementary Information.For compound 3a-3d, include purity assessment (e.g., HRMS, elemental analysis, or HPLC).For compound 3e-3k, include ^1^H NMR and clearly cite original literature if these compounds were previously reported.H NMR and clearly cite original literature if these compounds were previously reported.H NMR and clearly cite original literature if these compounds were previously reported.H NMR and clearly cite original literature if these compounds were previously reported.

**
Data Presentation and Formatting
**

Several technical issues require correction:

Reformat docking and ADMET tables for clarity.Correct typographical errors in tables (e.g., numeric formatting).Standardize units (μg/mL throughout).Correct formatting of log values (e.g., log Kp should not retain dimensional units).Revise figure captions for clarity and consistency (e.g., ensure compound numbers are correct).

**
Language and Editorial Revision
**

The manuscript contains multiple grammatical errors, tense inconsistencies, and awkward phrasing, all of which reduce clarity. Examples include incorrect verb forms (“used to synthesis”), missing articles (“progress of reaction”), and repetition in methods.

Reviewers' comments:

Reviewer's Responses to Questions

**Comments to the Author**

Reviewer #2: All comments have been addressed

2. Is the manuscript technically sound, and do the data support the conclusions?

Reviewer #2: Yes

3. Has the statistical analysis been performed appropriately and rigorously?

Reviewer #2: Yes

4. Have the authors made all data underlying the findings in their manuscript fully available?

Reviewer #2: Yes

5. Is the manuscript presented in an intelligible fashion and written in standard English?

Reviewer #2: Yes

Reviewer #2: The authors have almost carried out all the necessary corrections and hence the manuscript can be accepted for publication.

.

Reviewer #2: No

---

## [Author Response · Author response to Decision Letter 3]

9 Mar 2026

Dear Editor,

Thank you for the opportunity to revise our manuscript entitled "Synthesis of Substituted Biphenyls and In Vitro Evaluation of Antimicrobial and Anti-Biofilm Activities Supported by In Silico ADMET Prediction" (Manuscript ID: PONE-D-25-65844R2). We appreciate the thoughtful and constructive comments from the Editor, which have helped us significantly improve the quality of our work.

We've checked your submission and before we can proceed, we need you to address the following issues:

1. Please ensure that you refer to Figure 6 in your text as, if accepted, production will need this reference to link the reader to the figure.

Answer: Thank you for this note, we referred to figure 6in page 13

2. Please ensure that you refer to Table 5 in your text as, if accepted, production will need this reference to link the reader to the Table.

Answer: Thank you for this note, we referred to table 5 in page 13

---

## [Editor Report · Decision Letter 3]

11 Mar 2026

Dear Dr. Mohammed,

Thank you for submitting your manuscript to PLOS ONE. After careful consideration, we feel that it has merit but does not fully meet PLOS ONE’s publication criteria as it currently stands. Therefore, we invite you to submit a revised version of the manuscript that addresses the points raised during the review process.

We look forward to receiving your revised manuscript.

Kind regards,

Saki Raheem, PhD

Academic Editor

PLOS One

Journal Requirements:

Additional Editor Comments:

Thank you for revising the manuscript and addressing several of the previous comments. However, a detailed point-by-point response letter explaining how each editor’s comment was addressed was not provided. Please include a response document clearly indicating how each comment has been addressed and where the corresponding changes have been made in the revised manuscript.

Some issues also remain insufficiently addressed:

**1. Chemical characterization**

For compounds **3a–3d**, additional purity assessment should be provided. These compounds appear to be newly synthesized, yet only NMR and LC-MS data are reported. For newly reported compounds used in biological assays, it is generally expected to include at least one additional method confirming purity and molecular identity, such as HRMS or  elemental analysis data. , additional purity assessment should be provided. These compounds appear to be newly synthesized, yet only NMR and LC-MS data are reported. For newly reported compounds used in biological assays, it is generally expected to include at least one additional method confirming purity and molecular identity, such as HRMS or  elemental analysis data. , additional purity assessment should be provided. These compounds appear to be newly synthesized, yet only NMR and LC-MS data are reported. For newly reported compounds used in biological assays, it is generally expected to include at least one additional method confirming purity and molecular identity, such as HRMS or  elemental analysis data. , additional purity assessment should be provided. These compounds appear to be newly synthesized, yet only NMR and LC-MS data are reported. For newly reported compounds used in biological assays, it is generally expected to include at least one additional method confirming purity and molecular identity, such as HRMS or  elemental analysis data.

**2. Language and editorial revision**

The manuscript still contains numerous grammatical errors, tense inconsistencies, and awkward phrasing that affect readability. Examples include:

“Suzuki coupling reactions were used to synthesis of biphenyl compounds…”“software were used to predicte the 3i compound pharmacokinetic properties.”“softwers were used to calculate the physiochemical properties.”duplicated wording in the methods section (e.g., “mixed gently, and allowed to solidify, mixed gently, and allowed to solidify”).

I recommend that the manuscript undergo thorough language editing before resubmission. The authors may consider using professional language-editing services or grammar-checking tools such as Grammarly, Writefull, or similar writing software to improve grammar, syntax, and clarity.

**3. Consistency in compound characterization**

The manuscript states that the compounds were characterized using IR and NMR techniques; however, IR data are not reported in the experimental section. Please clarify this point or include the relevant spectroscopic data.

---

## [Author Response · Author response to Decision Letter 4]

14 Mar 2026

Dear Editor,

Thank you for the opportunity to revise our manuscript entitled "Synthesis of Substituted Biphenyls and In Vitro Evaluation of Antimicrobial and Anti-Biofilm Activities Supported by In Silico ADMET Prediction" (Manuscript ID: PONE-D-25-65844R2). We appreciate the thoughtful and constructive comments from the Editor, which have significantly improved the quality of our work.

1. Chemical characterization

For compounds 3a–3d, additional purity assessment should be provided. These compounds appear to be newly synthesized, yet only NMR and LC-MS data are reported. For newly reported compounds used in biological assays, it is generally expected to include at least one additional method confirming purity and molecular identity, such as HRMS or elemental analysis data.

Answer: "We thank the reviewer for this valuable suggestion. In accordance with the journal's requirements for newly synthesized compounds, we had performed CHN analysis for all compounds 3a–3d.

2. Language and editorial revision

The manuscript still contains numerous grammatical errors, tense inconsistencies, and awkward phrasing that affect readability. Examples include:

• “Suzuki coupling reactions were used to synthesis of biphenyl compounds…”

• “software were used to predicte the 3i compound pharmacokinetic properties.”

• “softwers were used to calculate the physiochemical properties.”

• duplicated wording in the methods section (e.g., “mixed gently, and allowed to solidify, mixed gently, and allowed to solidify”).

I recommend that the manuscript undergo thorough language editing before resubmission. The authors may consider using professional language-editing services or grammar-checking tools such as Grammarly, Write full, or similar writing software to improve grammar, syntax, and clarity.

Answer: We thank the reviewer for this valuable recommendation. We agree that improving the language quality is essential for the clarity of our manuscript. For this reason, we used the Grammarly program to edit the manuscript according to your suggestion

3. Consistency in compound characterization

The manuscript states that the compounds were characterized using IR and NMR techniques; however, IR data are not reported in the experimental section. Please clarify this point or include the relevant spectroscopic data.

Answer: We thank the reviewer for pointing out this omission. The IR and CHN data were reported in the experimental section

---

## [Editor Report · Decision Letter 4]

16 Mar 2026

Synthesis of Substituted Biphenyls and In Vitro Evaluation of Antimicrobial and Anti-Biofilm Activities Supported by In Silico ADMET Prediction

PONE-D-25-65844R4

Dear Dr. Mohammed,

We’re pleased to inform you that your manuscript has been judged scientifically suitable for publication and will be formally accepted for publication once it meets all outstanding technical requirements.

Kind regards,

Saki Raheem, PhD

Academic Editor

PLOS One